# Neotropical Studies on Hymenochaetaceae: Unveiling the Diversity and Endemicity of *Phellinotus*

**DOI:** 10.3390/jof8030216

**Published:** 2022-02-22

**Authors:** Carlos A. Salvador-Montoya, Samuel G. Elias, Orlando F. Popoff, Gerardo L. Robledo, Carlos Urcelay, Aristóteles Góes-Neto, Sebastián Martínez, Elisandro R. Drechsler-Santos

**Affiliations:** 1Laboratorio de Micología, Instituto de Botánica del Nordeste—IBONE, Universidad Nacional del Nordeste—UNNE, CC 209, Corrientes W3400, Argentina; popoff@exa.unne.edu.ar; 2Consejo Nacional de Investigaciones Científicas y Técnicas—CONICET, Godoy Cruz 2290, CABA, CP C1425FQB, Buenos Aires C1425, Argentina; gerardo.robledo@agro.unc.edu.ar (G.L.R.); curcelay@imbiv.unc.edu.ar (C.U.); 3Organización Juvenil “Hongos Perú”, Av. Ejército B-12, Cusco 08001, Peru; 4Departamento de Botânica, Centro de Ciências Biológicas, Campus Universitário, Universidade Federal de Santa Catarina—UFSC, Trindade, CEP: 88040-900, Florianópolis 88040-900, Brazil; sgelias@outlook.com; 5BioTecA3—Centro de Biotecnología Aplicada al Agro y Alimentos, Facultad de Ciencias Agropecuarias, Universidad Nacional de Córdoba, Félix Aldo Marrone 746—Planta Baja CC509—CP 5000, Ciudad Universitaria, Córdoba X5000, Argentina; 6Fundación Fungicosmos, Córdoba X5016, Argentina; 7Laboratorio de Micología, Instituto Multidisciplinario de Biología Vegetal, Universidad Nacional de Córdoba (UNC), CONICET, CC 495—CP 5000, Córdoba X5000, Argentina; 8Laboratório de Biologia Molecular e Computacional de Fungos (LBMCF), Departamento de Microbiologia, Instituto de Ciências Biológicas (ICB), Universidade Federal de Minas Gerais, Av. Antônio Carlos, 6627, Belo Horizonte 31270-901, Brazil; arigoesneto@gmail.com; 9Laboratorio de Patología Vegetal, Estación Experimental INIA Treinta y Tres, Instituto Nacional de Investigación Agropecuaria (INIA), Ruta 8 km 281, Treinta y Tres 33000, Uruguay; smartinez@inia.org.uy

**Keywords:** three new taxa, endemic fungi, phylogenetic lineages, SDTF species, taxonomy, xanthocroic polypores

## Abstract

*Phellinotus*, a neotropical genus of wood-decay fungi commonly found on living members of the Fabaceae family, was initially described as containing two species, *P*. *neoaridus* and *P*. *piptadeniae*. The members of this genus, along with six other well-established genera and some unresolved lineages, are the current representatives of the ‘phellinotus clade’. On the other hand, based on a two-loci phylogenetic analysis, some entities/lineages of the ‘phellinotus clade’ have been found in *Fomitiporella* s.l. In this work, we performed four-loci phylogenetic analyses and based on our results the genera of the ‘phellinotus clade’ are shown to be monophyletic groups. In addition to the natural groups confirmed as different genera, morphological revisions, phylogenetic relationships, and host distribution of different specimens resembling *P*. *neoaridus* and *P*. *piptadeniae* revealed three new species in the *Phellinotus* genus, referred to here as *P*. *magnoporatus*, *P*. *teixeirae* and *P*. *xerophyticus*. Furthermore, for *P*. *piptadeniae* a narrower species concept was adopted with redefined morphological characters and a more limited distribution range. Both *P*. *neoaridus* and *P*. *teixeirae* have a distribution range restricted to seasonally dry tropical forests in South America. Additionally, based on detailed morphological revisions *Phellinus* *badius*, *Phellinus* *resinaceus*, and *Phellinus* *scaber* are transferred to the *Phellinotus* genus. The geographic distribution and host range of the genus are then discussed.

## 1. Introduction

*Phellinotus* Drechsler-Santos, Robledo & Rajchenb. is a genus of wood-decay fungi characterized by perennial basidiomata with dark lines and a mycelial core, as well as a monomitic hyphal system in the context, a dimitic hyphal system in the tubes, and yellowish basidiospores with a flattened side that become chestnut brown in KOH solution [1]. The genus is defined using a morphological and phylogenetic approach. It is related to *Arambarria* Rajchenb. & Pildain, *Fomitiporella* Murrill s.str., *Fulvifomes* Murrill, *Inocutis* Fiasson & Niemelä, *Phylloporia* Murrill, and *Rajchenbergia* Salvador-Montoya, Popoff & Drechsler-Santos, as well as some other taxonomically unresolved lineages within the ‘phellinotus clade’ [1,2,3].

Phylogenetically, based on two-loci molecular data, previous studies have shown that the genera of the ‘phellinotus clade’ are monophyletic groups [1,2,3]. On the other hand, *Fomitiporella* is considered to be a broad category (*Fomitiporella* s.l.) and *Arambarria*, *Fomitiporella* s.str. *Phellinotus*, and *Rajchenbergia* are not considered natural groups [4,5]. Given these two contradictory positions on the lineages of the ‘phellinotus clade’, Pildain et al. [2] proposed three taxonomic scenarios. Based on morphological and phylogenetic analyses, the authors decided to consider all lineages of the ‘phellinotus clade’ different genera as the best option [2]. Pildain et al. [2] also mention that multilocus molecular data, as well as more taxa, could help to better understand the phylogenetic and taxonomic relationships in the ‘phellinotus clade’.

Currently, *Phellinotus* comprises two taxa that grow on species of Fabaceae Lindl., i.e., *P*. *piptadeniae* (Teixeira) Drechsler-Santos & Robledo and *P*. *neoaridus* Drechsler-Santos & Robledo [1,6,7]. *Phellinotus neoaridus* possesses black basidiomata with a rimose pileal surface, as well as dark lines and a mycelial core in the context. It grows on *Caesalpinia* L. spp. and *Cenostigma* Tul. spp. in the Caatinga biome, northeast Brazil [1]. *Phellinotus piptadeniae* develops basidiomata with a cracked to rimose grayish brown to olive gray pileal surface with a dark line in the context. This species grows mainly on living *Piptadenia gonoacantha* J.F. Macbr., but also on *Libidibia glabrata* (Kunth) Castellanos & G.P. Lewis, *Pithecollobium excelsum* (Kunth) Mart. and species of *Acacia* Mill., *Mimosa* L., *Piptadenia* Benth., *Pityrocarpa* Britton & Rose, and *Senegalia* Raf. It is distributed in the Caatinga biome, the southern part of Central Brazil, the Central Andes Coast, and the Misiones and Piedmont floristic groups (FG) of the seasonally dry tropical forests (SDTF) [1,6,7]. Elias et al. [7] consider *P*. *piptadeniae* to be a SDTF generalist species, with a broad geographic distribution in South America.

In this context, given that the macromorphology of *P. piptadeniae* is variable [6] and that high levels of substrate specificity in members of Hymenochaetaceae in South America have been proposed [8], we hypothesized that the current species concept does not recognize cryptic diversity in this species complex. To test this hypothesis through detailed morphological characterizations of basidiomata and molecular studies of specimens resembling *P*. *piptadeniae* and *P*. *neoaridus* from different hosts and regions of South America is the aim of this work. Furthermore, to assess species diversity in the genus and accurately identify entities, we aimed to perform morphological, molecular, and ecological (host and geographic distribution) analyses of collected specimens, as well as of type and reference specimens of *P*. *piptadeniae*, *P*. *neoaridus*, and taxa related to the *Phellinus rimosus* (Berk.) Pilát complex, namely *Ph*. *badius* (Cooke) G. Cunn., *Ph*. *resinaceus* Kotl. & Pouzar, and *Ph*. *scaber* (Berk.) M.J. Larsen.

## 2. Materials and Methods

### 2.1. Morphological Analyses

Specimens of *Phellinotus* were collected from different ecosystems in Argentina (2010), Brazil (2006–2016), Peru (2011–2012), and Uruguay (2011–2017). Types and reference material of *Phellinotus piptadeniae*, *P*. *neoaridus*, *Phellinus badius*, *Ph*. *resinaceus*, and *Ph*. *scaber* were morphologically analysed. The macro-morphological features (i.e., size, shape, and color) of the basidiomata (pileal surface, context, tubes, pores, and dissepiments) and the number of pores per mm were described. Color descriptions were based on the Munsell Color Company system [9] and Kornerup and Wanscher [10]. Micro-morphological observations were based on freehand sections. These sections were mounted in water and 5% KOH (*v*/*w*) solution, as well as Melzer’s reagent (to check for any dextrinoid or amyloid reactions) [11]. Microscopic measurements (*n* = 40) and drawings were made in 5% (*v*/*w*) KOH solution. To determine the size range of pores and microscopic structures (hyphae and basidiospores), 5% of the measurements were excluded from each end and are given in parentheses. The abbreviations ‘Q’ and ‘avg.’ represent the ratio of length/width [12] and average basidiospore measurements, respectively. Sections of the trama and context of the basidiomata were incubated in 5% (*v*/*w*) NaOH solution for 48 h at 40 °C and carefully dissected under a stereomicroscope to accurately describe the hyphal system [1,13].

### 2.2. Distribution

The distribution is presented following the floristic groups (FG) of the seasonally dry tropical forests (SDTF) proposed by Särkinen et al. [14] and DRYFLOR et al. [15], and the neotropical regionalization proposed by Morrone [16]. The maps were constructed with QGIS 3.0.3 “Girona” [17] from shape files provided by Särkinen et al. [14] and Löwenberg-Neto [18].

### 2.3. DNA Extraction, PCR Amplification, and Sequencing

Dried pieces of basidiomata were used to extract genomic DNA based on the protocol of Doyle and Doyle [19] modified by Góes-Neto et al. [20]. The *nITS* region was amplified with the primer pairs ITS1-F (5′-TCCGTAGGTGAACCTGCGG-3′) or ITS8-F (5′-AGTCGTAACAAGGTTTCCGTAGGTG-3′)/ITS4-R (5′-TCCTCCGCTTATTGATATGC-3′) or ITS6-R (5′-TTCCCGCTTCACTCGCAGT-3′) folowing the thermal conditions of 95 °C (2 min) for initial denaturation; 5 cycles of denaturation at 95 °C (30 s), annealing at 60 °C (30 s), extension at 72 °C (1 min); and final extension at 72 °C (10 min) [21,22]. *nLSU* amplification was performed using a LROR (5′-ACCCGCTGAACTTAAGC-3′)/LR7 (5′-TACTACCACCAAGATCT-3′) primer pair, following: initial denaturation at 94 °C (5 min); with 34 cycles of denaturation at 94 °C (1 min), annealing at 57 °C (1.5 min), extension at 72 °C (2 min); and final extension at 72 °C (10 min) [23]. A fragment between exons 4 and 8 of the translation elongation factor 1-alpha (TEF1-α) was amplified with primers 983F (5′-GCYCCYGGHCAYCGTGAYTTYAT-3′), 1567R (5′-ACHGTRCCRATACCACCSATCTT-3′), 1577F (5′-CARGAYGTBTACAAGATYGGTGG-3′), and 2212R (5′-CCRAACRGCRACRGTYYGTCTCAT-3′), following the thermal conditions of: 94 °C (2 min) for initial denaturation; 9 cycles of denaturation at 95 °C (30 s), slow-ramp annealing ranging from 66 to 75 °C with an increase of 1 °C/cycle (30 s), extension in 72 °C (1 min), and 36 cycles using the same thermal denaturation and extension conditions of the first nine cycles but using 56 °C (30 s) for annealing; and final extension at 72 °C (10 min) [24,25]. The most variable region of RNA polymerase II second largest subunit (RPB2) was amplified using primers bRPB2-5F (5′-GAYGAYMGWGATCAYTTYGG-3′), b6F (5′-TGGGGYATGGTNTGYCCYGC-3′), and b7.1R (5′-CCCATRGCYTGYTTMCCCATDGC-3′), following the thermal conditions of: 94 °C (5 min) for initial denaturation; 34 cycles of denaturation at 94 °C (1 min), annealing at 57 °C (1.5 min), extension at 72 °C (2 min); and final extension at 72 °C (10 min) [26,27,28]. Polymerase chain reaction (PCR) was performed with a total volume of 40 µL containing 20 µL of 2 × PCR Taq Master Mix (Applied Biological Material, Vancouver, BC, Canada), 1 µL of primer (10 pM), 1–2 µL of DNA, and q.s. sterile distilled water. Sanger sequencing was performed with a BigDye Terminator v.3.1 Cycle Sequencing Kit (Applied Biosystems, Foster, CA, USA) following the manufacturer’s instructions using the same primers. The sequencing was performed at FIOCRUZ-MG (Belo Horizonte, Brazil) and MACROGEN (Seoul, Korea). The new sequences were assembled and manually edited using Geneious v.6.1.8 [29].

### 2.4. Molecular Phylogenetic Analyses

Two combined datasets were developed concatenating the newly generated *nITS*, *nLSU*, TEF1-α, and RPB2 sequences of *Phellinotus* specimens and sequences of specimens/species of the ‘phellinotus clade’ retrieved from GenBank (NCBI) (Table 1). The sequences used in this study represented species and genera closely related to *Phellinotus* such as Drechsler-Santos et al. [1], Pildain et al. [2], Rajchenberg et al. [30], and Salvador-Montoya et al. [3,31,32]. The *nITS* and *nLSU* sequences were aligned separately in MAFFTv.7 [33]. The G–INS-i option was used to align sequences in the *nLSU* region and the Q–INS–i option was used for sequences in the *nITS* region. The TEF1-α and RPB2 sequences were aligned separately in MACSE (interface web: http://mbb.univ-montp2.fr/macse, accessed on 2 January 2022) [34]. In MEGA v.5.0 [35], the alignments were manually adjusted when necessary. Additionally, for protein-coding gene sequences (i.e., TEF1-α and RPB2), the assignment of codon positions was confirmed by translating nucleotide sequences into predicted amino acid sequences in MEGA v. 5.0. Potentially ambiguously aligned segments were detected with the software Gblocks 0.91b [36] for the *nITS* region.

Two distinct combined datasets were constructed: one with taxa the of the ‘phellinotus clade’ (with 36 specimens), and the other with species of *Phellnotus* (with 35 specimens). The two four-loci combined datasets (i.e., *nITS*, *nLSU*, TEF1-α and RPB2) were subdivided into 13 data partitions (ITS1, 5.8S, ITS2, LSU, TEF1-α-1st, -2nd, -3rd codon positions, TEF1-α introns, RPB2-1st, -2nd. and -3rd codon positions). The best-fit model of nucleotide evolution for the datasets was selected using AIC (Akaike Information Criterion) as implemented in jModelTest2 v.1.6 [37,38]. For phylogenetic inferences based on the combined dataset of the ‘phellinotus clade’ taxa, we selected *Sanghuangporus vaninii* (Ljub.) L.W. Zhou & Y.C. Dai, *Tropicoporus drechsleri* Salvador-Montoya & Popoff, and *Inonotus griseus* L.W. Zhou as outgroup species. For phylogenetic inferences from the combined dataset, species of *Phellinotus*, *T*. *drechsleri*, and *I*. *griseus* were selected as the outgroup species.

**Table 1 jof-08-00216-t001:** List of species, sources, localities, and GenBank accession numbers of taxa used in this study.

Species	Geographic Origin	Collection Reference	Substrate	GenBank Accession Number				Reference
				*nLSU*	*nITS*	TEF1-α	RPB2	
*Arambarria cognata*	Uruguay/Canelones	CGP473	*Dodonaea viscosa*	KY907687	KY907683	KY907675	-	[39]
*A*. *cognata*	Uruguay/Canelones	CGP474	*Dodonaea viscosa*	KY907692	KY907682	KY907676	-	[39]
*Fomitiporella austroasiana*	China/Hainan	Dai 16244	On fallen angiosperm trunk	MG657320	MG657328	-	-	[5]
*F*. *austroasiana*	China/Hainan	Dai 16168	On fallen angiosperm trunk	MG657321	MG657329	-	-	[5]
*F*. *caryophylli*	India	CBS 448.76	*Shorea robusta*	AY059021	AY558611	-	-	[40]
*F*. *chinensis*	China/Shaanxi	Cui 11230	*Quercus* sp.	KY693759	KX181309	KY693958	KY693910	[41]
*F*. *inermis*	USA/Pennsylvania	JV 1009/56	*Ilex mucronata*	KX181347	KX181306	-	-	[4]
*F*. *subinermis*	China/Hunan	Dai 15114	On root of angiosperm tree	KX181344	KX181308	-	-	[4]
*F*. *umbrinella*	Brazil/Santa Catarina	FLOR 51648	-	MK802941	MK802943	-	-	[3]
*F*. *umbrinella*	Brazil/Santa Catarina	FLOR 51649	-	MK802942	MK802944	-	-	[3]
*F*. *vietnamensis*	Vietnam/Lam Dong	Dai 18382	On angiosperm tree	MG657327	MG657333	-	-	[5]
*Fulvifomes elaeodendri*	South Africa	CMW47825	*Elaeodendron croceum*	MH599134	MH599094	MT108964	-	[42]
*F*. *squamosus*	Peru/Piura	USM 258361	*Acacia macracantha*	MF479266	MF479267	-	-	[31]
*I* *nocutis dryophilus*	USA	DLL2012-001	*Quercus alba*	KU139255	KU139186	-	KU139317	[43]
*I*. *jamaicensis*	USA/Arizona	RLG15819	*Quercus arizonica*	KY907703	KY907684	-	-	[44]
*I* *nonotus griseus*	China	Dai 13436	-	KX832925	KX674583	KY693959	KX364919	[45]
*Phellinotus magnoporatus*	Peru/Piura	USM 250523	*Ocotea aurantiodora*	MZ964981	MZ954859	OK000625	-	This study
*P*. *neoaridus*	Brazil/Pernambuco	URM 80362	*Caesalpinia* sp.	KM211286	KM211294	-	-	[1]
*P*. *neoaridus*	Brazil/Bahia	HUEFS 122186	*Cenostigma pyramidale*	MZ964976	-	-	-	This study
*P*. *neoaridus*	Brazil/Pernambuco	URM 77673	-	-	MZ954857	-	-	This study
*P*. *neoaridus*	Brazil/Bahia	URM 83203	-	MZ964977	MZ954858	-	-	This study
*P*. *neoaridus*	Brazil/Alagoas	URM 80579	*Caesalpinia* sp.	MZ964978	-	-	-	This study
*P*. *piptadeniae*	Brazil/Pernambuco	URM 80766	*Mimosa* sp.	KM211285	KM211293	-	-	[1]
*P*. *piptadeniae*	Brazil/Pernambuco	URM 80360	*Mimosa* sp.	KM211284	KM211292	-	-	[1]
*P*. *piptadeniae*	Brazil/Pernambuco	URM 80322	*Mimosa* sp.	KM211282	KM211290	-	-	[1]
*P*. *piptadeniae*	Brazil/Pernambuco	URM 80345	*Senegalia* sp.	KM211283	KM211291	-	-	[1]
*P*. *piptadeniae*	Brazil/Pernambuco	URM 80768	*Piptadenia* sp.	KM211281	KM211289	-	-	[1]
*P*. *piptadeniae*	Brazil/Santa Catarina	FLOR 51451	*Piptadenia gonoacantha*	MZ964964	MZ954839	-	-	This study
*P*. *piptadeniae*	Uruguay/Treinta y Tres	MVHC 5756	*Calliandra tweediei*	MZ964968	MZ954842	-	-	This study
*P*. *piptadeniae*	Uruguay/Treinta y Tres	MVHC 5754	*Calliandra tweediei*	MZ964967	MZ954843	-	-	This study
*P*. *piptadeniae*	Brazil/Santa Catarina	FLOR 63105	*Piptadenia gonoacantha*	MZ964971	MZ954847	OK000617	-	This study
*P*. *piptadeniae*	Brazil/Santa Catarina	FLOR 63111	*Piptadenia gonoacantha*	MZ964969	MZ954845	OK000618	-	This study
*P*. *piptadeniae*	Brazil/Rio de Janeiro	FLOR 62129	-	MZ964965	MZ954840	-	-	This study
*P*. *piptadeniae*	Uruguay/Treinta y Tres	MVHC 5561	*Calliandra tweediei*	KT266877	MZ954844	-	-	This study
*P*. *piptadeniae*	Brazil/Sao Paulo	FLOR 63627	*Piptadenia gonoacantha*	KP412282	KP412305	-	-	Genbank database
*P*. *piptadeniae*	Uruguay/Treinta y Tres	MVHC 5562	*Calliandra tweediei*	KT266878	KT266876	-	-	Genbank database
*P*. *piptadeniae*	Brazil/Santa Catarina	FLOR 63101	*Piptadenia gonoacantha*	MZ964970	MZ954846	OK000619	-	This study
*P*. *piptadeniae*	Brazil/Rio de Janeiro	FLOR 62132	-	MZ964966	MZ954841	-	-	This study
*P*. *piptadeniae*	Brazil/Santa Catarina	FLOR 39572	*Piptadenia gonoacantha*	-	MZ954848	-	-	This study
*P*. *teixeirae*	Brazil/Sergipe	URM 80403	*Piptadenia* sp.	-	MZ954849	-	-	This study
*P*. *teixeirae*	Argentina/Corrientes	CTES 515266	-	-	MZ954851	-	-	This study
*P*. *teixeirae*	Brazil/Pernambuco	URM 80636	*Pityrocarpa moniliformis*	-	MZ954850	-	-	This study
*P*. *teixeirae*	Brazil/Pernambuco	URM 80889	*Pityrocarpa moniliformis*	-	MZ954852	-	-	This study
*P*. *teixeirae*	Peru/Piura	USM 250528	*Pithecollobium excelsum*	MZ964972	MZ954853	-	-	This study
*P*. *teixeirae*	Peru/Piura	USM 258366	*Libidibia glabrata*	MZ964973	MZ954856	-	-	This study
*P*. *teixeirae*	Peru/Piura	USM 258362	*Libidibia glabrata*	MZ964975	MZ954854	OK000621	OK000626	This study
*P*. *teixeirae*	Peru/Piura	USM 278225	*Libidibia glabrata*	MZ964974	MZ954855	OK000622	OK000627	This study
*P*. *xerophyticus*	Argentina/Cordoba	CORD 3551	*Prosopis* sp.	MZ964979	-	OK000624	OK000629	This study
*P*. *xerophyticus*	Argentina/Cordoba	CORD 3552	*Prosopis* sp.	MZ964980	-	OK000623	OK000628	This study
*Phylloporia crataegi*	China	Dai 18133	*Crataegus* sp.	MH165865	MH151191	MH167431	MH161224	[46]
*P*. *pectinata*	Australia	Voucher 113	-	MH165867	MH151181	MH167421	MH161213	[46]
*P*. *pseudopectinata*	China	Cui 13749	Angiosperm	KX242356	MF410323	MH167429	MH161222	[47]
*P*. *rattanicola*	China	Dai 18232	On dead rattan	MG738806	MH151170	MH167414	MH161205	[48]
*Rajchenbergia mangrovei*	France/Guadeloupe	JV 1612/25-J	*Conocarpus erectus*	MG657331	MG657325	-	-	[3]
*R*. *pertenuis*	Brazil/Alagoas	PPT 111 (URM 91181)	On dead wood	MG806100	MG806101	-	-	[3]
*R*. *tenuissima*	China	Dai 12245	Angiosperm	KC999902	KC456242	-	-	[3]
*Sanghuangporus vaninii*	USA	DMR-95-1-T	*Populus tremuloides*	KU139258	KU139198	KU139380	KU139318	[43]
*T* *ropicoporus drechsleri*	Argentina/Chaco	CTES 570140	*Patagonula americana*	MG242444	MG242439	OK000620	-	[32]

Maximum likelihood (ML) and Bayesian inference (BI) analyses were applied to the combined datasets. ML searches were conducted with RaxMLv.8.1 [49], and the GTRGAMMA model was selected to search for the best-scored trees, with all other parameters estimated by the software. The analyses first involved 1000 ML independent searches, each one starting from one randomized stepwise addition parsimony tree. Only the best-scored ML tree was kept, and the reliability of the nodes was accessed through non-parametric Bootstrap (BS) pseudoreplicates under the same model, allowing the program to halt bootstrapping automatically with the autoMRE option. An additional alignment partition file to force RAxML software to search for a separate evolutionary model for each partition was used. BI analyses were carried out in MrBayes 3.2.6 [50] and implemented with two independent runs, each beginning from random trees with four simultaneous independent chains. A total of 2 × 10^7^ generations was carried out, sampling one tree every 1000th generation. The first 25% of the sampled trees were discarded as burn-in and checked by the convergence criterion (frequencies of average standard deviation of split <0.01) in Tracer v.1.6 [51], while the remaining ones were used to reconstruct a 50% majority-rule consensus tree and to estimate the Bayesian posterior probabilities (BPP) of the branches. jModelTest2 v.1.6, MrBayes 3.2.6, and RaxMLv.8.1 were used in CIPRES science gateway [52]. A node was strongly supported if it showed a BPP ≥ 0.9 and/or BS ≥ 90% while moderate support was considered if BPP ≥ 0.8 and/or BS ≥ 60%. For phylogenetic inferences of taxon based on the molecular data of the *nITS* region, we followed the recommendation of Jeewon and Hyde [53]. The topology from the ML tree is depicted, and the statistical values of ML and BI are indicated for each node. The alignments were deposited in TreeBASE under the accession number 28692 (http://purl.org/phylo/treebase/phylows/study/TB2:S28692, accessed on 2 January 2022).

## 3. Results

A total of 52 new sequences, i.e., 18 *nLSU*, 21 *nITS*, nine TEF1-α, and four RPB2 were generated. The four-loci combined dataset of the ‘phellinotus clade’ (ITS1+5.8S+ITS2+LSU+TEF1-α-1st, -2nd, -3rd codon positions+ TEF1-α introns+RPB2 -1st, -2nd. and -3rd codon positions) includes 36 entries representing nine genera (*Arambarria*, *Fomitiporella* s.str., *Fulvifomes*, *Inocutis*, *Phellinotus*, *Phylloporia* and *Rajchenbergia*), resulting in an alignment with 3877 characters, of which 2055 are constant, 1642 variable, and 1124 parsimony informative. The bootstopping criteria from the ML analysis stopped after 300 replicates. The best evolutionary model estimated for each dataset was TrN+I+G, TrN+I+G, K80, HKY+G, TrN+I, TPM2uf+I, GTR+G, TPM1uf+I, TIM1+I, TIM3+I, TIM2+G, TIM3+G, and TPM2uf+G for LSU, ITS1, 5.8S, ITS2, TEF1-α-1st codon, TEF1-α-2nd codon, TEF1-α-3rd codon, TEF1-α-4th intron, TEF1-α-5th intron, TEF1-α-6th intron, RPB2-1st codon, RPB2-2nd codon, and RPB2-3rd codon, respectively.

The general topology recovered in the phylogenetic inferences based on the four-loci combined dataset of the ‘phellinotus clade’ (Figure 1) is congruent with previous studies [1,2,3] and seven genera were recovered as monophyletic lineages with strong support in the ‘phellinotus clade’: *Arambarria* (BS = 100/BPP = 1), *Inocutis* (BS = 100/BPP = 1), *Fomitiporella* s.str. (BS = 100/BPP = 1), *Fulvifomes* (BS = 100/BPP = 1), *Phellinotus* (BS = 90/BPP = 1), *Phylloporia* (BS = 100/BPP = 1), and *Rajchenbergia* (BS = 97/BPP = 1), besides the reminiscent groups of *Fomitiporella* s.l.

The four-loci combined dataset of *Phellinotus* (ITS1+5.8S+ITS2+LSU+TEF1-α-1st, -2nd, -3rd codon positions + TEF1-α introns + RPB2-1st, -2nd, and -3rd codon positions) included 35 sequences representing specimens of *Phellinotus*, resulting in an alignment with 3566 characters, of which 2508 were constant sites, 931 variable, and 360 parsimony informative. The bootstopping criteria from the ML analysis stopped after 1000 replicates. The best evolutionary model estimated for the alignment was TIM2+G, K80+G, TIM2ef+I, TPM3uf+G, F81+I, HKY+I, TPM2uf, HKY+I, K80+G, TIM3+G, TPM2uf+I, TIM1, and TPM2 for LSU, ITS1, 5.8S, ITS2, TEF1-α-1st codon, TEF1-α-2nd codon, TEF1-α-3rd codon, TEF1-α-4th intron, TEF1-α-5th intron, TEF1-α-6th intron, RPB2-1st codon, RPB2-2nd codon, and RPB2-3rd codon, respectively.

Phylogenetic inferences from the four-loci combined dataset of *Phellinotus* (Figure 2) revelead three new taxa in the genus, named *P*. *magnoporatus* sp. nov. (described below), *P*. *teixeirae* sp. nov. (described below), and *P*. *xerophyticus* sp. nov. (described below). The sequences of *P*. *teixeirae* sp. nov. formed a monophyletic group strongly supported by (BS = 100/BPP = 1) and closely related to *P*. *piptadeniae* (BS = 100/BPP = 1), and both of these are related to *P*. *xerophyticus* sp. nov. (BS = 100/BPP = 1), *P*. *neoaridus* (BS = 99/BPP = 1), and *P*. *magnoporatus* sp. nov.

In accordance with the results of this study based on morphological and molecular data, *P*. *magnoporatus* sp. nov., *P*. *teixeirae* sp. nov. and *P*. *xerophyticus* sp. nov. are identified as new taxa in the *Phellinotus* genus. Additionally, based on detailed morphology, *Phellinus badius*, *Phellinus resinaceus*, and *Phellinus scaber* are transferred to the *Phellinotus* genus.

### Taxonomy

***Phellinotus badius*** (Berk. ex Cooke) Salvador-Montoya, Popoff & Drechsler-Santos comb. nov. (Figure 3D–K and Figure 4A–D).

≡*Polyporus badius* Berk., Annals and Magazine of Natural History 7: 453 (1841) (nom. ileg., non. *P*. *badius* [Pers.] Schw.).

≡*Fomes badius* Berk. ex Cooke, Grevillea 14(69): 18 (1885).

≡*Microporus badius* (Berk. ex Cooke) Kuntze, Revisio generum plantarum 3(2): 495 (1898).

≡*Scindalma badium* (Berk. ex Cooke) Kuntze, Revisio generum plantarum 3(2): 518 (1898).

≡*Trametes badia* (Berk. ex Cooke) Pat., Essai taxonomique sur les familles et les genres des Hyménomycètes: 93 (1900).

≡*Coriolopsis badia* (Berk. ex Cooke) Murrill, Bulletin of the Torrey Botanical Club 34: 466 (1907).

≡*Polysticus badius* (Berk. ex Cooke) Lloyd, Mycol. Notes 65: 1038 (1921).

≡*Fomes rimosus* var. *badius* (Berk. ex Cooke) Rick, Iheringia 7: 205 (1960).

≡*Phellinus badius* (Berk. ex Cooke) G. Cunn., Bulletin of the New Zealand Department of Industrial Research 164: 233 (1965).

≡*Fomitiporella badia* (Berk. ex Cooke) Teixeira, Revista Brasileira de Botânica 15(2): 125 (1992).

Mycobank: MB 840993

*Typification*: NORTH AMERICA: *Polyporus badius* n. sp. No. 6 suberoso lignosi, Dr. Richardson (K–M 199720!, lectotype).

*Description*: Basidioma perennial, pileate, sessile, broadly attached, ungulate, solitary, up to 70 mm long, 60 mm wide, and 30 mm thick, woody hard; pileal surface glabrous, rough and dark reddish brown (HUE 5YR, 3/2), slightly cracked, concentrically sulcate with a few deep fissures; margin entire, round to obtuse, thick, pubescent and dark brown (HUE 7,5YR, 4/6); pore surface flat and dark brown (HUE 7,5YR, 3/2), pores rounded, regular, 4–5/mm, (160–)190–350(–370) µm diam, dissepiments entire, (40–)50–160(–170) µm thick; context up to 15 mm thick, with a dark line near the pileal surface, non zonate and dark brown (HUE 7,5YR, 4/6), tubes indistinctly stratified, up to 10 mm long, dark brown (HUE 7,5YR, 4/6).

Hyphal system monomitic in the context and dimitic in the tubes; context dominated by generative hyphae, (3–)4–5(–6.5) µm diam, branched, thin-walled to gradually thick-walled, occasionally septate, trama of tubes with thin to slightly thick-walled generative hyphae, simple septate, branched, and unbranched skeletal hyphae, thick-walled with a visible to solid lumen, (150–)170–479(–595) µm long × (3.5–)4–5(–5.5) µm diam (L avg. = 334.93 µm, W avg. = 4.5 µm), tapering to the apex where the wall is thin and three to four adventitious septa are present, setae absent, cystidioles absent, abundant quadrangular or rhomboid crystals in the hymenium and trama of tubes, basidia not observed, basidiospores broadly ellipsoid to ellipsoid, with a flattened side, 6–7(–8) × (4.5–)5–6 µm (L avg. = 7.2 µm, W avg. = 5.3 µm), Q = 1.25–1.50 (Q avg. = 1.36), thick-walled, smooth, pale brownish yellow in water, turning chestnut to ferruginous brown in KOH, IKI-.

*Habitat and distribution*: The substrata and distribution data of species are still unknown [54]. According to Larsen [55], this is from North America; however, Cunningham [56] believes that this taxon belongs to the West Indies.

*Notes*: This species, described by Berkeley [57], is considered a species of *Phellinus* by its perennial basidioma and dimitic hyphal system (*Phellinus* s.l.) [54,55,56,58,59]. However, we revised the lectotype of species (K–M 199720) and observed the slightly cracked pileal surface, the hymenophore with 4–5 pores/mm, the context with a dark line, a monomitic hyphal system in the context and dimitic in the tubes, absence of setae, and broadly ellipsoid to ellipsoid basidiospores (7–8 × 5–6 µm) with a flattened side that turns darker in KOH solution (Table 2, Figure 3D–K and Figure 4A–D). In this case, this taxon is transferred to *Phellinotus* because it fits the morphological concept of the genus [1]. *Phellinotus badius* resembles *P*. *neoaridus* and *P*. *piptadeniae*, but these last two have smaller basidiospores (5.5–6.5 × 4–5 μm in *P*. *neaoridus*; 5–5.5 × 3.5–4.5 μm in *P*. *piptadeniae*) (Table 2). Additionally, *P*. *neoaridus* has basidiomata with a rimose pileal surface and a mycelial core in the context [1], whereas *P*. *piptadeniae* has basidiomata with a lobulate pileal surface in well-developed specimens (Table 2, Figure 5D,G,H).

***Phellinotus magnoporatus*** Salvador-Montoya & Drechsler-Santos, sp. nov. (Figure 3A–C and Figure 4E–G)

Mycobank: MB 840994

*Typification*: PERU. Piura: Las Lomas, Parque Nacional Cerros de Amotape, 4.194472S, 80.461320W, on living tree of *Ocotea aurantiodora* (Ruiz & Pav.) Mez., 6 December 2011, C. A. Salvador-Montoya 372 (USM 250523!, holotype; FLOR 51897!, isotype).

*Diagnosis*: Basidioma perennial, ungulate, pileal surface slightly cracked at the base of basidioma, margin round, hymenophore poroid (1–2/mm). Context with a mycelial core. Hyphal system monomitic in the context and dimitic in the tubes. Basidiospores broadly ellipsoid to ellipsoid with a flattened side, 4.5–5.5 × 4–4.4 µm, thick-walled, yellowish, chestnut to ferruginous in KOH, on living tree of *O*. *aurantiodora*.

*Etymology*: *magnoporatus*, refers to the large pores.

*Description*: Basidioma perennial, pileate, sessile, broadly attached, solitary, triquetrous to ungulate, up to 53 mm long, 80 mm wide, and 45 mm thick, woody hard, pileal surface glabrous, rough and very dark grayish brown (HUE 10YR, 3/2), concentrically zoned with a few deep fissures when well developed, margin entire, round, thick, pubescent, and dark yellowish brown (HUE 10YR, 4/6), pore surface flat and dark brown (HUE7, 5YR, 3/4), pores rounded, some elongated, regular, 1–2(–3)/mm, (300–)350–600(–650) µm diam, dissepiments entire, (100–)110–250(–270) µm thick, context up to 37 mm thick at the base in well-developed specimens, with a mycelial core in the base of basidioma, non-zonate and yellowish brown (HUE 10YR, 5/8), tubes indistinctly stratified, up to 15 mm long, yellowish brown (HUE 10YR, 5/8), with whitish mycelia strands usually filling the old tubes.

Hyphal system monomitic in the context and dimitic in the tubes, context dominated by generative hyphae, (2–)3–5.5(–6) µm diam, branched, thin-walled to gradually thick-walled, occasionally septate, trama of tubes with thin to slightly thick-walled generative hyphae, simple septate, branched, and unbranched skeletal hyphae, thick-walled with a visible to solid lumen, (130–)193–448(–497) µm long × (3.5–)3–4(–4.5) µm diam (L avg. = 293.2 µm, W avg. = 3.7 µm), tapering to the apex where the wall is almost thin and three to four adventitious septa are present, setae absent, cystidioles absent, without any crystals in the trama of tubes and dissepiments, basidia not observed, basidiospores broadly ellipsoid to ellipsoid, with a flattened side, (4–)4.5–5.5(–6) × (3–)4–4.5 µm (L avg. = 5.0 µm, W avg. = 4.0 µm), Q = 1.10–1.50 (Q avg. = 1.24), thick-walled, smooth, pale yellow in water, turning chestnut to ferruginous brown in KOH, IKI-.

*Habitat and distribution*: In the Central Andes Coast floristic group of SDTFs in South America, basidiomata of *P*. *magnoporatus* are found on living trees of *Ocotea aurantiodora*.

*Notes*: *Phellinotus magnoporatus* is characterized by the ungulate basidioma with 1–2 pores/mm and the context with a mycelial core (Table 2, Figure 3A–C). This species resembles *P*. *neoaridus*. Nevertheless, *P*. *neoaridus* has rimose basidiomata, hymenophore with 4–5 pores/mm, and a context with dark lines (Table 2, Figure 6A–C). Furthermore, *P*. *neoaridus* is a parasitic polypore on *Caesalpinea* and *Cenostigma* species (trees of Fabaceae) in the Caatinga biome of northern Brazil [1]. *Phellinotus magnoporatus* can be confused with *P*. *badius*. Both species have ungulate basidiomata with a slightly cracked pileal surface, a hymenophore with larger pores, and a hyphal system that is monomitic in the context and dimitic in the tubes. However, *P*. *badius* has a dark line in the context (Table 2, Figure 3G,H).

***Phellinotus piptadeniae*** (Teixeira) Drechsler-Santos & Robledo, 2016. (Figure 4 and Figure 7A–C).

≡*Phellinus piptadeniae* Teixeira, Bragantia 10(4): 118. 1950.

≡*Fomitiporella piptadeniae* (Teixeira) Teixeira, Revista Brasileira de Botânica 15(2): 126. 1992.

Mycobank: MB805902

*Typification*: BRAZIL. São Paulo: Parque Horto Florestal da Cantareira, Serviço Florestal do Estado, on living tree of *Piptadenia communis* Benth. (current name = *P*. *gonoacantha*), 14 March 1949, A. R. Teixeira s.n. (F 15071!, isotype).

*Description*: Basidioma perennial, pileate, sessile, broadly attached, solitary, applanate, triquetrous to ungulate, up to 300 mm long, 250 mm wide, and 150 mm thick, woody hard, pileal surface at first pubescent and brown (5E7–5E8), soon glabrous and greyish olive (1E5–1E6), when young wavy and concentrically sulcate, when mature becomes lobulate and cracked, concentric and radially sulcate, mostly fine sulci but some coarse and deep, margin entire, round, thick, pubescent and yellowish brown (5D8) to brown (5E7–5E8), pore surface flat and dark brown (6F6–6F7), pores rounded, regular, (3–)4–6(–7)/mm, (100–)120–300(–320) μm diam, dissepiments entire, (20–)30–190(–200) μm thick, context up to 60 mm thick, with a sinuous and dark line, zoned and light brown (6D8), tubes distinctly stratified with context layers among the strata of tubes, up to 100 mm long, light brown (6D8) to brown (6E8).

Hyphal system monomitic in the context and dimitic in the tubes, context dominated by generative hyphae, (1.5–)2.5–9(–10) µm diam, branched, thin-walled to gradually thick-walled, occasionally septate, trama of tubes with thin to slightly thick-walled generative hyphae, simple septate, branched, and unbranched skeletal hyphae, thick-walled with a visible to solid lumen, (165–)175–636(–754) μm long × (3–)3.5–9(–10) µm diam (L avg. = 385.6 µm, W avg. = 5.9 µm), tapering to the apex where the wall is almost thin and three to four adventitious septa are present, setae absent, cystidioles absent, without any crystals in the trama of tubes and dissepiments, basidia not observed, basidiospores broadly ellipsoid to ellipsoid, with a flattened side, (4–)4.5–5.5(–6) × (3–)3.5–4.5(–5) µm (L avg. = 5.2 µm, W avg. = 4.0 µm), Q = 1.10–1.50 (Q avg. = 1.33), thick-walled, smooth, pale yellow in water, turning chestnut to ferruginous brown in KOH, IKI-.

*Habitat and distribution*: Basidiomata of this parasitic polypore are found predominately on living trees of *Piptadenia gonoacantha*. It also grows on *Calliandra tweediei* Benth. and *Eugenia uruguayensis* Cambess. (Myrtaceae Juss.), besides *Mimosa*, *Senegalia* and *Piptadenia* species. This polypore is distributed from northern Brazil to central Uruguay, in the Caatinga, Cerrado, Atlantic, Parana Forest and Pampean provinces, and in the SDTFs of eastern South America.

*Notes*: According to Salvador-Montoya et al. [6], Drechsler-Santos et al. [1], and Elias et al. [7], *P*. *piptadeniae* has applanate to triquetrous basidiomata with cracked to rimose reddish yellow to grayish black pileal surface, a hymenophore with 4–6 pores/mm, a context with a dark line, tubes distinctly stratified, skeletal hyphae restricted to the trama of the tubes, and broadly ellipsoid to ellipsoid with a flattened side basidiospores that turn darker in KOH solution. Additionally, this is a SDTFs generalist species that is found predominately on living trees of *P*. *gonoacantha* (recurrent host). However, based on a phylogenetic and morphological reassessment of various specimens for this study, we revealed two entities within this taxon (*P*. *piptadeniae* s.l.). One entity possesses a rimose and dark grayish brown pileal surface in mature basidiomata and grows on living trees of Fabaceae in different FG of SDTFs in South America (described as *P*. *teixierae* sp. nov. below) (Table 2, Figure 1 and Figure 2). The second entity has a lobulate, cracked, olive gray pileal surface with deep concentric furrows delimiting wide lobes in mature specimens and grows predominately on living trees of *P*. *gonoacantha* in different types of vegetation (Table 2, Figure 5D,G,H). In this context, this last species is considered as *P*. *piptadeniae* s.str. in this work. *Phellinotus piptadeniae* resembles *P*. *neoaridus*. However, *P*. *neoaridus* has a rimose pileal surface, a mycelial core in the context, tubes indistinctly stratified and slightly larger basidiospores (5.5–6.5 × 4–5) (Table 2, Figure 6A–C). Furthermore, *P*. *neoaridus* is a parasitic polypore on *Caesalpinia* and *Cenostigma* species in the Caatinga biome of Brazil [1].

*Specimens examined*: BRAZIL. São Paulo: Campinas, Bosque dos Jequitibás, on living tree of *P*. *communis* (current name = *P*. *gonoacantha*), 12 October 1943, A. R. Teixeira & P. R. Santos s.n. (IAC 4365!, paratype); Município de Botucatu, trilha Ecológica Casa da Natureza, Fazenda Experimental do Lajeado, on living tree of *P*. *gonoacantha*, 11 January 2013, M. Fernandes MF32 (FLOR 63616!); *id*. M. Fernandes MF34 (FLOR 63617!); *id*. M. Fernandes MF31 (FLOR 63615!); *ibid*. on living tree *P*. *gonoacantha*, 30 January 2013, M. Fernandes MF7 (FLOR 19926!); *id*. M. Fernandes MF8 (FLOR 30457!); *ibid*. on living tree of *P*. *gonoacantha*, 20 February 2014, M. Fernandes MF38 (FLOR 63621!); *ibid*. on living tree of *P*. *gonoacantha*, 22 February 2014, M. Fernandes MF43 (FLOR 63626!); *ibid*. on living tree of *P*. *gonoacantha*, 26 February 2014, M. Fernandes MF44 (FLOR 63627!); *ibid*. on living tree of *P*. *gonoacantha*, 4 July 2013, M. Fernandes MF26 (FLOR 39430!); *id*. M. Fernandes MF27 (FLOR 51449!); *ibid*. on living tree of *P*. *gonoacantha*, 4 July 2013, M. Fernandes MF29 (FLOR 51450!); Caraguatatuba, Parque Nacional da Serra do Mar, Núcleo Caraguatatuba, 23.594167S, 45.420278W, on living tree of *P*. *gonoacantha*, 19 January 2016, S. Galvão-Elias SGE220 (FLOR 63114!); *id*. S. Galvão-Elias SGE221 (FLOR 63115!); *id*. S. Galvão-Elias SGE222 (FLOR 63116!); *id*. S. Galvão-Elias SGE224 (FLOR 63118!); *id*. S. Galvão-Elias SGE226 (FLOR 63120!); *ibid*. on living tree of *P*. *gonoacantha*, 20 January 2016, S. Galvão-Elias SGE242 (FLOR 63121!); *id*. S. Galvão-Elias SGE243 (FLOR 63122!); *id*. S. Galvão-Elias SGE244 (FLOR 63123!); *id*. S. Galvão-Elias SGE245 (FLOR 63124!); Santa Catarina: Florianópolis, Campus Universitário/UFSC, on living tree of *P*. *gonoacantha*, 25 January 2011, M. A. Borba-Silva MABS106 (FLOR 39571!); *ibid*. on living tree of *P*. *gonoacantha*, 14 April 2011, M. A. Borba-Silva MABS135 (FLOR 39572!); *id*. M. A. Borba-Silva MABS136 (FLOR 39573!); Criciúma, Parque Municipal José Milanese, 28.713056S, 49.378333W, on living tree of *P*. *gonoacantha*, 06 September 2014, S. Galvão-Elias SGE93 (FLOR 63099!); Criciúma, Parque Municipal Morro do Céu, 28.713056S, 49.378333W, on living tree of *P*. *gonoacantha*, 16 May 2015, S. Galvão-Elias SGE110 (FLOR 63100!); *id*. S. Galvão-Elias SGE111 (FLOR 63101!); *id*. S. Galvão-Elias SGE113 (FLOR 63102!); *ibid*. on living tree of *P*. *gonoacantha*, 17 May 2015, S. Galvão-Elias SGE114 (FLOR 63103!); *id*. S. Galvão-Elias SGE115 (FLOR 63104!); *id*. S. Galvão-Elias SGE117 (FLOR 63105!); Florianópolis, Moro da Lagoa, 27.604444S, 48.491389W, on living tree of *P*. *gonoacantha*, 15 September 2015, S. Galvão-Elias SGE120 (FLOR 63106!); *id*. S. Galvão-Elias SGE124 (FLOR 63108!); *id*. S. Galvão-Elias SGE125 (FLOR 63109!); *id*. S. Glavão-Elias SGE126 (FLOR 63110!); *id*. S. Galvão-Elias SGE127 (FLOR 63111!); Tubarão, Fazenda Lunard, trilha do rio, on trunk of *P*. *gonoacantha*, 14 November 2012, A. G. Silva-Filho AGS48 (FLOR 51451!); Pernambuco: Estação Experimental do IPA, Caruarú, on *Mimosa* sp., 10 December 2008, E. R. Drechsler-Santos 109PE (URM 80322!); *ibid*. on *Senegalia* sp., 10 December 2008, E. R. Drechsler-Santos 110PE (URM 80345!); *ibid*. Serra Talhada, on *Piptadenia* sp., 5 March 2009, E. R. Drechsler- Santos 139PE (URM 80768!); Rio de Janeiro: Parque Nacional da Tijuca, trilha do corcovado, 23 November 2014, M. A. Reck MAR836 (FLOR 62129!); Parque Nacional da Tijuca, 26 November 2014, M. A. Reck MAR920 (FLOR 62132!); *ibid*. on living tree of *P*. *gonoacantha*, 09 April 2016, S. Galvão-Elias SGE348 (FLOR 63092!); *id*. S. Galvão-Elias SGE349 (FLOR 63089!); *id*. S. Galvão-Elias SGE350 (FLOR 63090!); *id*. S. Galvão-Elias SGE351B (FLOR 63096!); *id*. S. Galvão-Elias SGE352 (FLOR 63094!); *ibid*. on living tree of *P*. *gonoacantha*, 15 April 2016, S. Galvão-Elias SGE354 (FLOR 63091!); *ibid*. on living tree of *P*. *gonoacantha*, 08 April 2016, S. Galvão-Elias SGE253B (FLOR 63093!); Distrito Federal, Brasilia: Parque Olhos d’ Água, 15.742778S, 47.885278W, on living tree of *P*. *gonoacantha*, 01 July 2016, S. Galvão-Elias SGE385 (FLOR 63068!); *id*. S. Galvão-Elias SGE387 (FLOR 63070!); *id*. S. Galvão-Elias SGE388 (FLOR 63071!); *id*. S. Galvão-Elias SGE389 (FLOR 63072!); *id*. S. Galvão-Elias SGE390 (FLOR 63073!); Paraná: Maringá, Parque do Ingá, 23.425833S, 51.931944W, on living tree of *P*. *gonoacantha*, 14 August 2016, S. Galvão-Elias SGE417 (FLOR 63082!); *id*. S. Galvão-Elias SGE406A (FLOR 63085!); *id*. S. Galvão-Elias SGE410A (FLOR 63086!); *id*. S. Galvão-Elias SGE420A (FLOR 63086!). URUGUAY. Treinta y Tres: Arroyo Yerbal y R98, 33.216667S, 54.395278W, on living tree of *Calliandra tweediei*, 11 June 2011, S. Martínez Y1 (MVHC 5561!); *id*. S. Martínez Y3 (MVHC 5562!); *ibid*. on dead trunk of Fabaceae, 11 June 2011, S. Martínez Y2 (MVHC 5748!); *ibid*. on *Eugenia uruguayensis*, 11 June 2011, S. Martínez Y18 (MVHC 5749!); Arroyo Yerbal, Playa Calera, 33.197500S, 54.385278W, on dead trunk of Fabaceae, 02 July 2011, S. Martínez PY1 (MVHC 5563!); *ibid*. on living tree of *C*. *tweediei*, 22 April 2017, S. Martínez Calera 2 (MVHC 5753!); *id*. S. Martínez Calera 3 (MVHC 5754!); Río Olimar, frente Villa Passano, 33.276667S, 53.885278W, on living tree of *C*. *tweediei*, 20 November 2011, S. Martínez VP1 (MVHC 5750!); Río Olimar, Unidad Experimental de Paso de la Laguna, 33.279722S, 54.158611W, on living tree of *C*. *tweediei*, 22 November 2012, S. Martínez EP1 (MVHC 5751!); Río Olimar Chico y R19, Palo a Pique, 33.265278S, 54.532778W, on living tree of *C*. *tweediei*, 18 June 2011, S. Martínez OC6 (MVHC 5752!); Ciudad, Río Olimar, 33.239722S, 54.407222W, on living tree of *C*. *tweediei*, 30 April 2019, S. Martínez 1907 (MVHC 5755!); *id*. S. Martínez 1908 (MVHC 5756!); *id*. S. Martínez 1909 (MVHC 5757!); *id*. S. Martínez 1910 (MVHC 5758!); *id*. S. Martínez 1911 (MVHC 5759!).

***Phellinotus resinaceus*** (Kotl. & Pouzar) Salvador-Montoya & Drechsler-Santos comb. nov. (Figure 4H–J and Figure 8H–P)

≡*Phellinus resinaceus* Kotl. & Pouzar, Folia Geobotanica et Phytotaxonomica 14 (3): 261 (1979).

Mycobank: MB 840995

*Typification*: PAPUA NEW GUINEA. Badili: Port Moresby, near Methodist Church, on *Eucalyptus papuana* F. Muell., 23 August 1964, S. Wuai 4252 (K–M: 180663!, holotype).

*Description*: Basidioma perennial, pileate, sessile, broadly attached, solitary, ungulate, up to 41 mm long, 60 mm wide, and 53 mm thick, woody hard, pileal surface glabrous, brown (5E4–5F4), with some resinous, dark, and glossy substance, cracked, concentrically, and radially sulcate, mostly fine sulci but some coarse and deep, margin entire, round, thick, pubescent, and golden brown (5D7), pore surface flat to slightly convex, dark blond (5D4) to brown (5E4), pores rounded, regular, (2–)3–4(–5)/mm, (220–)230–400(–420) μm diam; dissepiments entire, (40–)50–120(–130) μm thick, context up to 13 mm thick, with a mycelial core at the base of the basidioma, with some resinous substance in the form of granules and thin layers, non-zonate and brown (6D8–6E8), tubes indistinctly stratified, up to 40 mm long, brown (6D8–6E8), with whitish mycelia strands usually filling the old tubes.

Hyphal system monomitic in the context and dimitic in the tubes, context dominated by generative hyphae, (4–)5–7.5(–8) μm diam, branched, thin-walled to gradually thick-walled, occasionally septate, trama of tubes with thin to slightly thick-walled generative hyphae, simple septate, branched, and unbranched skeletal hyphae, thick-walled with a visible to solid lumen, (217–)239–575(–587) μm long × (3–)4–5(–6) μm diam (L avg. = 428.5 μm, W avg. = 4.6 μm), tapering to the apex where the wall is almost thin and three to four adventitious septa are present, setae absent, cystidioles absent, without any crystals in the trama of tubes and dissepiments, basidia not observed, basidiospores broadly ellipsoid to ellipsoid, with a flattened side, (6–)6.5–7.5(8) × (4.5–)5–6(–6.5) μm (L avg. = 7.1 μm, W avg. = 5.6 μm), Q = 1.17–1.44 (Q avg. = 1.27), thick-walled, smooth, pale yellow in water, turning chestnut to ferruginous brown in KOH, IKI-.

*Habitat and distribution*: This polypore is distributed in Papua New Guinea and Australia, growing on living tree of *E*. *papuana*.

*Notes*: This taxon is considered a species of *Phellinus* (*Phellinus* s.l.) [54,60]. However, we observed the type specimens of species (K–M: 180663, holotype; PRM 671088, paratype), which had a mycelial core in the context at the base of the basidiomata, a monomitic hyphal system in the context and dimitic in the tubes, and basidiospores (5.5–7.5 × 5–6 μm) broadly ellipsoid to ellipsoid with a flattened side that turn darker in KOH solution (Table 2, Figure 7H–J, and Figure 4H–P). This taxon is transferred to *Phellinotus* because it fits the morphological concept of the genus [1]. *Phellinotus resinaceus* resembles *P*. *magnoporatus* and *P*. *badius*. However, *P*. *magnoporatus* has hymenophore with larger pores (1–2/mm) and smaller basidisopores (4.5–5.5 × 4–4.5 μm), and *P*. *badius* has a dark line in the context (Table 2, Figure 3 and Figure 4).

*Specimen examined*: AUSTRALIA. New South Wales: Sydney, 29 March 1953, K. Caval s.n. (PRM 671088!, paratype).

***Phellinotus scaber*** (Berk.) Salvador-Montoya, Drechsler-Santos & Popoff comb. nov. (Figure 9 and Figure 10A–C).

≡*Polyporus igniarius* var. *scaber* Berk., Annals and Magazine of Natural History 3: 324 (1839).

≡*Fomes scaber* (Berk.) Lloyd, Synopsis of the genus Fomes (7): 249 (1915).

≡*Phellinus scaber* (Berk.) M.J. Larsen, Mycotaxon 37: 356 (1990).

Mycobank: MB 840996

*Typification*: AUSTRALIA. Tasmania (Van Diemen’s Land): V. D. Lawrence s.n. (K–M: 180666!, lectotype).

*Description*: Basidioma perennial, pileate, sessile, broadly attached, solitary, triquetrous to ungulate, up to 86 mm long, 54 mm wide, and 49 mm thick, woody hard, pileal surface at first pubescent and dark brown (HUE 7,5YR, 4/6), soon glabrous and black (HUE 7,5YR, N2/0) to very dark gray (HUE 7,5YR, N3/0), rough when young with a few deep fissures, mature specimens become rimose, concentric and radially sulcate, with deep sulci, margin entire, round, thick, pubescent, and yellow (HUE 2,5YR, 8/8) to strong brown (HUE 7,4YR, 5/8), pore surface flat to concave and yellow (HUE 5Y, 7/6) to olive brown (HUE 5Y, 4/3), pores rounded, regular, 3–4(–5)/mm, (160–)170–520(–570) μm diam; dissepiments entire, (30–)40–250(–260) μm thick, context up to 15 mm thick, dark brown (HUE 7,5YR, 4/4) to very dark brown (HUE 7,5YR, 3/4), with dark lines, non-zonate, tubes indistinctly stratified, up to 37 mm long, dark brown (HUE 7,5YR, 4/4), with whitish mycelia strands usually filling the old tubes.

Hyphal system monomitic in the context and dimitic in the tubes, context dominated by generative hyphae, (2.5–)3–8(–9) μm diam, branched, thin-walled to gradually thick-walled, occasionally septate, trama of tubes with thin to slightly thick-walled generative hyphae, simple septate, branched, and unbranched skeletal hyphae, thick-walled with a visible to solid lumen, (76–)87–717(–782) μm long × (4–)6–8(–9) μm diam (L avg. = 343.7 μm, W avg. = 5.8 μm), tapering to the apex where the wall is almost thin and three to four adventitious septa are present, setae absent, cystidioles absent, without any crystals in the trama of the tubes and dissepiments, basidia not observed, basidiospores broadly ellipsoid to oblong, occasionally subglobose, with a flattened side, (4.5–)5.5–8(–8.5) × (3–)4–6(–6.5) μm (L avg. = 6.3 μm, W avg. = 4.6 μm), Q = 1.09–1.86 (Q avg. = 1.35), thick-walled, smooth, pale yellow in water, turning chestnut to ferruginous brown in KOH, IKI-.

*Habitat and distribution*: The perennial basidiomata are found on living trees of *Acacia* (*A*. *baileyana* F. Muell. and *A*. *melanoxylon* R. Br.), *Allocasuarina* L.A.S. Johnson, *Casuarina* J.R. Forst. & G. Forst. and *Eucalyptus* L’Her. (*E*. *baueriana* Schauer and *E*. *camaldulensis* Dehnh.) species. This taxon has a wide distribution in Australia.

*Notes*: This taxon, described by Berkeley [61] as *Polyporus igniarius* var. *scaber*, was considered a synonym of *Phellinus rimosus* (Berk.) Pilát (current name = *Fulvifomes rimosus* (Berk.) Fiasson & Niemelä) on account of its perennial basidiomata with rimose pileal surface and dimitic hyphal system (*Phellinus* s.l.) [55,59,62]. We revised the type specimen of *Polyporus igniarius* var. *scaber*, and other material from Australia previously determined as *P*. *rimosus*. All material showed triquetrous to ungulate basidiomata with rimose and dark gray to black pileal surface, flat to concave, yellowish to olive, hymenophore with 3–4 pores/mm, the context with dark lines and a monomitic hyphal system, a dimitic hyphal system in the tubes, without setae, and broadly ellipsoid to oblong with a flattened side basidiospores (5.5–8 × 4–6 μm) that turn darker in KOH solution (Table 2, Figure 9 and Figure 10A–C). Accordingly, this species is transferred to *Phellinotus* as the best option because it fits into the morphological concept of the genus [1] as well. *Phellinotus scaber* resembles *P*. *neoaridus*. However, *P*. *neoaridus* has a reddish yellow to dark brown and flat to convex hymenophore (Table 2, Figure 6A–C). Furthermore, *P*. *neoaridus* is a parasitic polypore on *Caesalpinia* and *Cenostigma* species in the Caatinga biome of Brazil [1].

*Specimens examined*: AUSTRALIA. Northern Territory: Tephrina Gorge, Ross River area, Macdonell Ranges, 23.516667S, 134.400000E, on living tree of *Eucalyptus camaldulensis*, 1 June 1974, J. H. Willis s.n. (MEL 229175!); Tasmania: Trevallyn State Recreation Area (near Launceston), 41.433333S, 147.100000E, on fallen stem of *Allocasuarina* sp., 12 September 2003, G. M. Gates & D. A. Ratkowsky WR119 (MEL 2359296!); Victoria: Midlands, Long Forest Nature Conservation Reserve, 37.650000S, 144.500000E, on pollarded *E*. *baueriana*, 8 August 2004, N. H. Sinnott 3413 (MEL 2264869); *ibid*. on living tree of *Eucalyptus* sp., 28 August 2004, N. H. Sinnott 3418 (MEL 2264870!); Gippsland Plain, Providence Ponds rest area, Princes Highway (WSW of Bairnsdale), Gippsland, 37.916667S, 147.266667E, on dead branch of *Acacia baileyana*, 24 September 2006, N. H. Sinnott 3681 (MEL 2305369!); Eastern Highlands, next to Rubicon River Road, 3.5 Km E of Rubicon township, 37.315556S, 145.852778E, on living tree of *Acacia melanoxylon*, 16 June 1996, K. Macfarlane 28 (MEL 2032383!); Western Australia: Roe, Hippos Yawn, Wave Rock, just east of Hyden, 32.444167S, 118.903611E, on living tree of *Casuarina* sp., 26 November 1996, J. H. Ross 3860 (MEL 2041343!).

***Phellinotus teixeirae*** Salvador-Montoya, Elias & Drechsler-Santos, sp. nov. (Figure 6D–L and Figure 7D–F).

Mycobank: MB 840997

*Typification*: PERU. Piura: Las Lomas, Parque Nacional Cerros de Amotape, 4.194472S, 80.461320W, on living tree of *Pithecellobium excelsum* (Kunth) Mart., 7 December 2011, C. A. Salvador-Montoya 377 (USM 250528! holotype, FLOR 7554! isotype).

*Diagnosis*: Basidioma perennial, triquetrous, pileal surface rimose, margin round, hymenophore poroid (4–6/mm). Context with dark lines. Hyphal system monomitic in the context and dimitic in the tubes. Basidiospores broadly ellipsoid to ellipsoid with a flattened side (4.5–6.5 × 3.5–5.5 µm), thick-walled, yellowish, chestnut to ferruginous in KOH, on living tree of Fabaceae in SDTFs.

*Etymology*: *teixeirae*, in honor of Dr. Alcides Ribeiro Teixeira for his valuable contribution to mycology in Brazil.

*Description*: Basidioma perennial, pileate, sessile, applanate to ungulate, solitary, occasionally imbricate, up to 135 mm long, 65 mm wide, and 58 mm thick, woody hard, pileal surface first pubescent and dark brown (HUE 7,5YR, 4/6), soon glabrous and dark grayish brown (HUE 2.5Y, 4/2), when young concentrically wavy with fissures, later becoming concentric and radially coarsely slightly cracked, when fully mature concentric and radially cracked, with deep sulci, margin entire, thick, round, pubescent and dark yellowish brown (HUE 10YR, 4/6) to dark grayish brown (HUE 2.5Y, 4/2), pore surface flat and dark brown (HUE 7,5YR, 3/4), pores rounded, regular, (3–)4–6(–7) per mm, (110–)130–350(–390) µm diam, dissepiments entire, (30–)40–130(–140) µm thick; context up to 8 mm thick in mature specimens, dark yellowish brown (HUE 10YR, 4/6), zonate, with one or two dark lines, tubes indistinctly stratified, up to 40 mm long, dark brown (HUE 7,5YR, 3/4).

Hyphal system monomitic in the context and dimitic in the trama of tubes, context dominated by generative hyphae, (1.5–)2–8(–9) µm diam, simple septate, branched, thin-walled, gradually thick-walled, occasionally portions with only a few septa observed; trama of tubes dimitic with thin- to slightly thick-walled generative hyphae, simple septate, branched, and unbranched skeletal hyphae, thick-walled with a visible lumen to almost solid, (135–)138–630(–675) µm long × (3–)3.5–6.5(–7) µm diam (L avg. = 384.5 µm, W avg. = 4.73 µm), tapering to the apex where the wall is almost thin and three to four adventitious septa are present, setae absent, cystidioles absent, without any crystals in the trama of tubes and dissepiments, basidia not observed, basidiospores broadly ellipsoid to ellipsoid, with a flattened side, (4–)4.5–6.5(–7) × (3–)3.5–5.5(–6) µm (L avg. = 5.6 µm, W avg. = 4.2 µm), Q = 1.10–1.57 (Q avg. = 1.33), thick-walled, smooth, pale yellow in water, turning chestnut to ferruginous brown in KOH, IKI-.

*Habitat and distribution*: The perennial basidiomata are found growing on living trees of *Pithecellobium excelsum*, *Libidibia glabrata* and *Pityrocarpa moniliformis* (Benth.) Luckow & R.W. Jobson, besides *Acacia* and *Piptadenia* species. This taxon is distributed in the Caatinga, Misiones, Piedmont and Central Andes Coast FC of SDTFs in South America.

*Notes*: *Phellinotus teixeirae* is characterized by applanate to ungulate basidiomata with rimose and dark grayish brown pileal surface, the hymenophore with 4–6 pores/mm, dark lines in the context, a monomitic hyphal system in the context and dimitic in the tubes, absence of setae, and basidiospores (4.5–6.5 × 3.5–5.5 µm) broadly ellipsoid to ellipsoid with a flattened side that turn darker in KOH solution (Table 2, Figure 6D–L and Figure 7D–F). The young basidiomata of this species resemble specimens of *P*. *piptadeniae* (Figure 5E). However, the mature specimens of *P*. *piptadeniae* have a lobulate and cracked olive gray pileal surface, with deep concentric furrows and wide lobes (Table 2, Figure 5D,G,H). In addition, *P*. *piptadeniae* is a recurrent parasite of *Piptadenia gonoacanhta*, with distribution from northern Brazil to Uruguay in various ecosystems in the east of South America. *Phellinotus teixeirae* can be confused with *P*. *neoaridus* and *P*. *scaber*. Both species have dark lines in the context. However, *P*. *neoaridus* has a rimose and black pileal surface, besides a mycelial core in the context (Table 2, Figure 6A–C). Additionally, *P*. *neoaridus* is found on living trees of *Caesalpinea* and *Cenostigma* species in the Caatinga biome of Brazil [1]. *Phellinotus scaber* differs by having an olive brown concave hymenophore, larger pores (3–4/mm) and basidiospores (5.5–8 × 4–6 μm), and a distribution in Australia, growing on *Acacia*, *Allocasuarina*, *Casuarina*, and *Eucalyptus* species (Table 2, Figure 9).

*Specimens examined*: PERU. Piura: Las Lomas, Parque Nacional Cerros de Amotape, 4.312185S, 80.546895W, on living tree of *Libidibia glabrata*, 28 August 2012, C. A. Salvador-Montoya 454b (USM 278225!, duplicate FLOR 16944!); *ibid*. on living tree of *Libidibia glabrata*, 29 August 2012, C. A. Salvador-Montoya 457b (USM 258362!, duplicate FLOR 16945!); *id*. C. A. Salvador-Montoya 461b (USM 258366!, duplicate FLOR 16946!). BRAZIL. Pernambuco: Buíque, Parque Nacional do Catimbau, Quixadeira/Morro do cachorro, 8.409722S, 37.248333W, on living tree of *Pityrocarpa moniliformis*, 30 October 2007, E. R. Drechsler-Santos et al. DS257 (URM80889!); Buíque, Parque Nacional do Catimbau, Trilha das Torres/Igrejinha, 8.571389S, 37.24611W, on living tree of *Pityrocarpa moniliformis*, 08 December 2006, E. R. Drechsler-Santos et al. DS108 (URM 80636!); Sergipe: Niterói, 9.755S × 37.4625W, on *Piptadenia* sp., 16 June 2008, E. R. Drechsler-Santos et al. DS51SE (URM 80403!). ARGENTINA. Corrientes: Itatí, Scorza Cué, 27.278150S, 58.246331W, 11 January 1988, O. Popoff et Dichtiar OP345 (CTES 515266!); ARGENTINA. Anta: Salta, Parque Nacional El Rey, 24.694444S, 64.611056W, on trunk of *Acacia* sp., 24 March 2007, O. Popoff et al. OP4566 (CTES 569014!).

*Additional specimens examined*: *Phellinotus neoaridus*: BRAZIL. Pernambuco: Serra Talhada, Estação Experimental do IPA, 7.891389S, 38.304722W, on living caatingueira tree (*Caesalpinia* sp.), 09 December 2008, Drechsler-Santos DS105PE (URM 80362, holotype, FLOR 53152!, isotype); *ibid*. on living tree of *Caesalpinia* sp., 12 September 2007, E. R. Drechsler-Santos DS131PE (URM 80299!); *ibid*. on living tree of *Caesalpinia* sp., 5 March 2009, E. R. Drechsler-Santos DS519PE (URM 80769!); Barra da Jangada, Jobatão dos Guararapes, 9.546944S, 37.557500W, on living tree of Fabaceae, September 2003, Silva GT s/n (URM 77673!); Sergipe: Poço Redondo, Trilha de Angicos, 9.3808S, 37.4042W, on living tree of *Caesalpinia* sp. 14 June 2008, E. R. Drechsler-Santos DS437 (URM 80419!); Niterói, 9.4523S, 37.2756W, on living tree of *Caesalpinia* sp., 16 June 2008, E. R. Drechsler-Santos DS438 (URM 80422!); Porto da Folha, 9.544S, 37.1616W, on living tree of *Caesalpinia* sp., 15 June 2008, E. R. Drechsler-Santos DS456 (URM 80577!); Alagoas: São José de Tapera, Fazenda do Sr. Rudá, 9.3249S, 37.3327W, on living tree of *Caesalpinia* sp., 17 June 2008, E. R. Drechsler-Santos DS457 (URM 80579!); *ibid*. on living tree of *Caesalpinia* sp., 9 February 2009, E. R. Drechsler-Santos DS435 (URM 80411!); Bahia: Curaçá, 21 February 2011, C. R. S. Lira 141 (URM83203!); *ibid*. 23 February 2011, C. R. S. Lira 173 (URM 83194!); Ipirá, Fazenda Nova Favela, 12.176667S, 39.776667W, 23 July 2004, A. Góes-Neto 1043 (HUEFS 102940!); *ibid*. on living tree of *Cenostigma pyramidale* (Tul.) Gagnon & G.P. Lewis, 17 August 2006, A. Góes-Neto 1496 (HUEFS 118024!); *id*. A. Góes-Neto & L. César 1 (HUEFS 122186!); Senhor do Bonfim, Serra do Santana e Fazenda Passaginha, 10.371111S, 40.181111W, on living tree, 11 September 2006, J. R. T. Vasconcellos-Neto 196 (HUEFS 133885!); Ceará: Reriutaba, Serrote de Boqueirão, 4.020278S, 40.645278W, on living tree of *Caesalpinia* sp., 15 June 2007, E. R. Drechsler-Santos DS255 (URM 80847!); *ibid*. on living tree of *Caesalpinia* sp., E. R. Drechsler-Santos DS195 (URM 80536!); Paraíba: Catolé do Rocha, Fazenda Sta Idalina, on dead trunk, 25 February 1980, M. A. Sousa & C. A. B. Miranda 842 (JPB 3859!); Areia, Mata Pau-Ferro, on dead trunk, 19 September 1984, M. A. Sousa 2039 (JPB 8470!); Município do Pombal, on living tree of “Jurema-Preta”, 18 January 1986, M. A. Sousa 2180 (JPB 9287!).

***Phellinotus xerophyticus*** Robledo, Urcelay & Drechsler-Santos, sp. nov. (Figure 8A–G and Figure 10D–F).

Mycobank: MB 840998

*Typification*: ARGENTINA. Córdoba: Pocho, Reserva Provincial Chacani, 31.341667S, 65.468889W, on living tree of *Prosopis* sp., 18 July 2010, G. L. Robledo & E. R. Drechsler-Santos 2099 (CORD 3551!, holotype).

*Diagnosis*: Basidioma perennial, ungulate, pileal surface rimose, margin round, hymenophore poroid (3–4/mm). Context with a dark line. Hyphal system monomitic in the context and dimitic in the tubes. Basidiospores broadly ellipsoid to ellipsoid with a flattened side (6.5–7 × 5–5.5 µm), thick-walled, yellowish, chesnut to ferruginous in KOH, on living *Prosopis* spp.

*Etymology*: *xerophyticus*, refers to the dry environment where the species is found.

*Description*: Basidioma perennial, pileate, sessile, triquetrous to ungulate, solitary, up to 72 mm long, 75 mm wide, and 64 mm thick, woody hard, pileal surface rimose, concentric and radially sulcate, with deep sulci, dark gray (HUE 10YR, 3/1) to black (HUE 2,5Y, 2/0) in well-developed specimens, margin entire, thick, round, glabrous and black (HUE 2,5Y, 2/0), pore surface flat to convex and dark yellowish brown (HUE 10YR, 3/4), pores rounded, regular, 3–4(–5) per mm, (210–)240–360(–420) µm diam, dissepiments entire, (50–)60–121(–170) µm thick; context up to 4 mm thick in well-developed specimens, dark yellowish brown (HUE 10YR, 4/6), with a dark line not zonate, tubes indistinctly stratified, up to 50 mm long, dark yellowish brown (HUE 10YR, 3/6), with yellowish mycelia strands usually filling the old tubes.

Hyphal system monomitic in the context and dimitic in the trama of tubes, context dominated by generative hyphae (3.5–)4–5.5(–6) µm diam, regularly septate, branched, thin-walled, gradually thick-walled, occasionally portions where few septa are observed; trama of tubes dimitic with thin- to slightly thick-walled generative hyphae, simple septate, branched, and unbranched skeletal hyphae, thick-walled with a visible lumen to almost solid, (203–)217–463(–562) µm long × (2–)3–5(–5.5) µm diam (L avg. = 316.4 µm, W avg. = 3.7 µm), tapering to the apex where the wall is almost thin and three to four adventitious septa are present, setae absent, cystidioles absent, without any crystals in the trama of tubes and dissepiments, basidia not observed, basidiospores broadly ellipsoid to ellipsoid, occasionally subglobose, with a flattened side, (6–)6.5–7(–7.5) × (4.5–)5–5.5 (–6) µm (L avg. = 6.8 µm, W avg. = 5.3 µm), Q = 1.20–1.40 (Q avg. = 1.30), thick-walled, smooth, pale yellow in water, turning chestnut to ferruginous brown in KOH, IKI-.

*Habit and distribution*: In the Chaquean province of South America, basidiomata are found on living trees of *Prosopis* sp.

*Notes*: *Phellinotus xerophyticus* was previously identified as *Phellinus rimosus* [63]. It is characterized by ungulate and rimose basidiomata with hymenophore having 3–4 pores/mm and the context with a dark line (Table 2, Figure 8A–G). This species resembles *P*. *neoaridus*. Nevertheless, *P*. *neoaridus* has hymenophore with slightly smaller pores (4–5/mm) and basdiospores (5.5–6.5 × 4–5 µm), and the context has a mycelial core (Table 2, Figure 6A–C). Furthermore, *P*. *neoaridus* is a parasitic polypore on *Caesalpinea* and *Cenostigma* species in the Caatinga biome of northern Brazil [1]. *Phellinotus xerophyticus* can be confused with *P*. *teixeirae* and *P*. *scaber*. All three species have ungulate and rimose basidiomata. However, *P*. *teixeirae* has hymenophore with slightly smaller pores (4–6/mm) and has slightly smaller basidiospores (4.5–6.5 × 3.5–5.5 µm), whereas *P*. *scaber* has a concave hymenophore and dark lines in the context (Table 2, Figure 6D–L and Figure 9). Additionally, *P*. *teixeirae* is found growing on different species of Fabaceae in different FG of SDTFs of South America, and *P*. *scaber* grows on living trees of *Acacia*, *Allocasuarina*, *Casuarina* and *Eucalyptus* species in Australia.

*Additional specimens examined*: ARGENTINA. Córdoba: Pocho, Reserva Provincial Chacani, 31.341667S, 65.468889W, on living tree of *Prosopis* sp., 18 July 2010, G. L. Robledo & E. R. Drechsler-Santos 2098 (CORD 3552!).

## 4. Discussion

### 4.1. Taxonomic Status of Phellinotus badius, P. resinaceus and P. scaber

*Phellinotus badius*, *P*. *resinaceus*, and *P*. *scaber* are herein presented in a correct taxonomic status. Historically, *P*. *badius* and *P*. *scaber* had taxonomic or nomenclatural problems. Different authors transferred these two taxa to different genera [55,56,62,64,65,66,67,68,69] and morphologically they were related to the *Phellinus rimosus* complex [55,59]. Finally, *Phellinotus badius*, *P*. *resinaceus* and *P*. *scaber* were considered within the wide morphological concept of *Phellinus* (*Phellinus* s.l.) for many decades [54,55,56,58,59].

Given these facts, we revised the type specimens of *P*. *badius*, *P*. *resinaceus*, and *P*. *scaber* in this work. Based on our observations, the types of these three species have characters that are present in taxa of the ‘phellinotus clade’, such as a dark line, a mycelial core, a monomitic hyphal system in the context, a dimitic hyphal system in the tubes, basidiomata without setae, and colored basidiospores with a flattened side that turn darker in KOH solution [1]. Based on the revision of the literature, Kotlaba and Pouzar [59] and Larsen [54] observed yellowish basidiospores that turn chestnut brown in KOH solution in the type specimens of *P*. *badius* and *P*. *scaber*. Therefore, in accordance with the results of this work and our reinterpretation of the literature, we transferred these three species to *Phellinotus*. However, molecular analyses are suggested to confirm their phylogenetic position in this genus.

In the ‘phellinotus clade’, *Arambarria*, *Fulvifomes*, *Inocutis*, *Phylloporia*, and *Rajchenbergia* also contain species with a dark line or a mycelial core in the context. However, *Arambarria*, *Inocutis*, and *Rajchenbergia* have a monomitic hyphal system in the basidiomata, *Fulvifomes* has a monomitic to dimitic hyphal system, and *Phylloporia* mainly has yellowish basidiospores in KOH solution [1,3,30,39,44,48,70].

### 4.2. Morphological, Host and Geographic Distribution Patterns in Species of Phellinotus

The species of *Phellinotus* are presented here with a narrower morphological concept and a redefined, more limited, distribution range. In Hymenochaetaceae, *Phellinotus* is considered a genus with a neotropical distribution [1]. Based on our results, the geographic distribution of *Phellinotus* is expanded due to the addition of *P*. *badius*, *P*. *resinaceus*, and *P*. *scaber* to this genus. *Phellinotus resinaceus* and *P*. *scaber* have a geographic distribution in the Australasian region. *Phellinotus resinaceus* grows on *Eucalyptus papuana* (Myrtaceae) and is characterized by the presence of resinous substances in the basidiomata and a mycelial core in the context (Table 2, Figure 8H–M). On the other hand, *P*. *scaber* is found on living members of the family Myrtaceae, Casuarinaceae R. Br. and Fabaceae, and is characterized by a rimose pileal surface and the presence of dark lines in the context (Table 2, Figure 9A–K). In the case of *P*. *badius*, a species characterized by a cracked pileal surface and a dark line in the context (Table 2, Figure 3D–H), its type locality and substrata remain unknown. However, based on our interpretation of the literature, this taxon is distributed in North and Central America, as well as the Caribbean region [54,56]. Considering these facts, *Phellinotus* has a geographic distribution in the tropical and subtropical climatic zones of America and Australia, growing on living members of different genera of angiosperms.

In South America, the geographic distribution pattern of *P*. *piptadeniae* has been studied [1,6,7]. This species was described by Teixeira [71] as a parasitic polypore on *P*. *communis* (current name = *P*. *gonoacantha*), from the Atlantic Forest of the Brazilian state of São Paulo. Sixty years later, Drechsler-Santos et al. [1] extended its distribution to the Caatinga domain in northern Brazil, where it was found on living trees of the species *Piptadenia* Benth. Then, Salvador-Montoya et al. [6] revealed that the populations of this species showed variable morphology and had a disjunct distribution in different patches of SDTF in South America, within the limits of the Atlantic Forests domain in Brazil and northwestern Peru, growing on living tree species belonging to different genera of Fabaceae such as *Libidibia* Schltdl., *Mimosa*, *Pithecellobium* Mart., *Pityrocarpa*, and *Senegalia*. Recently, Elias et al. [7] showed that *P*. *piptadeniae* is a SDTF generalist species with a wide geographic distribution in South America that decays several hosts (with *P*. *gonoacantha* as a recurrent host).

Given this evidence, the wide taxonomic concept of *P*. *piptadeniae* has been questioned. Based on the results of this study, two taxa are revealed: *P*. *piptadeniae* s.str. and *P*. *teixeirae* (Figure 1 and Figure 2). Morphologically, *P*. *piptadeniae* s.str. is characterized by a cracked lobulate with concentric deep furrows delimiting wide lobes, olive gray pileal surface in well-developed specimens (Table 2, Figure 5A–I). Furthermore, this parasitic polypore is distributed both in dry and wet forest in eastern South America, within the limits of the Caatinga, Cerrado, Atlantic, Parana Forest, and Pampean provinces and SDTFs (Figure 11B), growing on living tree species of Fabaceae (ex. *C*. *tweediei* and *P*. *gonoacantha* (a recurrent host)) and Myrtaceae (ex. *E*. *uruguayensis*). Therefore, the evidence supports that this taxon is not a specialist, as suggested by Elias et al. [7].

*Phellinotus teixeirae* differs from *P*. *piptadeniae* s.str. mainly in terms of its rimose and dark grayish brown pileal surface (Table 2, Figure 6D–I). The morphological patterns of the pileal surface have been documented to distinguish species in the Hymenochaetaceae; for example, *Fulvifomes squamosus* Salvador-Montoya & Drechsler-Santos and *F*. *cedrelae* (Murrill) Murrill with *F*. *robiniae* (Murrill) Murrill, and the latter with *F*. *rimosus* [31,59]. Furthermore, *P*. *teixeirae* is distributed within the limits of SDTF in South America (Figure 11C), growing on living tree species of Fabaceae such as *Pithecellobium excelsum*, *Libidibia glabrata*, and *Pityrocarpa moniliformis*. These three tree species are distributed within the limits of SDTF, which makes them related to some degree [14,72,73,74]. *Libidibia glabrata* is a common species in the inter-andean seasonally dry tropical forest and in the lowland of the equatorial seasonally dry tropical forest [73], *P*. *excelsum* is found in the seasonally dry forests both in the Andean valleys in northern Peru and southern Ecuador and on the pacific coast of Ecuador and northern Peru [72], and *P*. *moniliformis* grows in the Caatinga, as well as the Atlantic Forests domain, in southeastern and northeastern Brazil [75,76,77]. Additionally, Särkinen et al. [14] mentioned that *P*. *moniliformis* is a SDTF habitat specialist. Therefore, unlike *P*. *piptadeniae* s.str., *P*. *teixeirae* is considered a SDTF host specialist in this work. The SDTF specialist species are entities that restrictedly grow in SDTF habitats [14,78].

*Phellinotus neoaridus* is a parasitic polypore that morphologically resembles *P*. *piptadeniae* s.str. and *P*. *teixeirae*. However, *P*. *neoaridus* differs from them by having a black rimose pileal surface and a mycelial core in the context (Table 2, Figure 6A–C). Furthermore, *P*. *neoaridus* is widely distributed in the Brazilian semiarid region (Caatinga dry woodlands), growing on living trees of *Caesalpinia* spp. and *Cenostigma pyramidale* [1]. *Cenostigma pyramidale* is an endemic species in the Caatinga domain and is widely distributed in northeastern Brazil [79,80,81,82]. Based on our results and interpretation of the literature, all records of *P*. *neoaridus* are found within the limits of SDTF in northeastern Brazil (Figure 11A). Therefore, we consider *P*. *neoaridus* a SDTF host specialist, as suggested for *P*. *teixeirae*. Accordingly, both *P*. *neoaridus* and *P*. *teixeirae* are considered endemic polypore species from SDTF in this study. In the Hymenochaetaceae, some taxa are treated as endemic species. For example, *Coltricia africana* Masuka & Ryvarden is endemic to Africa [83]. In South America, *Inonotus venezuelicus* Ryvarden, *Fomitiporia tabaquilio* (Urcelay, Robledo & Rajchenb.) Decock & Robledo, *Phellinus daedaliformis* J.E. Wright & Blumenf., and *P*. *uncisetus* Robledo, Urcelay & Rajchenb. are host specialist species (endemics) of *Polylepis* Ruiz & Pav. forests in the Córdoba mountains [8,84].

With respect to *P*. *magnoporatus* and *P*. *xerophyticus*, both species are only registered from their type localities in South America (Figure 11A). Both species have large pores (1–2 pores/mm in *P*. *magnoporatus*, 3–4 pores/mm in *P*. *xerophyticus*). However, *P*. *magnoporatus* is characterized by a cracked pileal surface and a context with a mycelial core (Table 2, Figure 3A–C) and grows on *O*. *aurantiodora* in the Central Andes Coast floristic group of SDTFs, while *P*. *xerophyticus* has a rimose pileal surface and a dark line in the context (Table 2, Figure 8A–E) and grows on species of *Prosopis* L. in the Chaquean province. Further surveys and studies would help shed light on their distributional ranges.

### 4.3. Classification System Status in the ‘Phellinotus Clade’

Previous studies, based on multilocus molecular data, have been performed to delimit the lineages/entities in the Hymenochaetaceae family [85,86,87,88,89,90,91]. Pildain et al. [2] mentioned that the incorporation of more loci, as well as taxa, into phylogenetic analyses of the ‘phellinotus clade’ could help us to better understand the taxonomic inferences in this group. At that time, three taxonomic scenarios were proposed based on phylogenetic inferences of a two-loci combined dataset (*nITS* and *nLSU*). The first taxonomic scenario assumes that different lineages in the group of the ‘phellinotus clade’ are different genera (i.e., *Arambarria*, *Inocutis*, *Fomitiporella* sensu Pildain et al. [2], *Fulvifomes*, and *Phylloporia*), while the second scenario treats the taxa of the ‘phellinotus clade’ as a unique genus, *Fomitiporella*. The third taxonomic scenario keeps *Phellinotus* and *Inocutis* as independent genera and groups *Arambarria* under *Fomitiporella* [2]. The first taxonomic scenario was accepted as the best option despite the morphological and phylogenetic variability.

As suggested by Pildain et al. [2], in this work we conducted a phylogenetic analysis with a four-loci combined dataset (*nITS*, *nLSU*, TEF1-α, and RPB2) and based on our results we agree with the first scenario of classification proposed by Pildain et al. [2]. Despite our phylogenetic inferences suggesting that the ‘phellinotus clade’ genera (i.e., *Arambarria*, *Inocutis*, *Fomitiporella* s.str., *Fulvifomes*, *Phylloporia*, *Rajchenbergia* and other non-taxonomically treated lineages in *Fomitiporella* s.l.) is a monophyletic group with high support (Figure 1), we agree that it is a classification hypothesis. In the future, following further research on natural groups at the genera level on Hymenochaetaceae, this will become more evident. We would also like to reinforce the need to conduct more sampling with the inclusion of specimens from different ecosystems around the world, as well as analyses based on more molecular markers.


**
*Key to the known species of Phellinotus*
**
1. Basidiomata with resinous substances in the pileal surface and the context *..................................................................Phellinotus resinaceus*1′. Basidiomata without resinous substances in the pileal surface and the context..............................................................................................22. Hymenophore yellow to olive brown, flat to concave *..............................................................................................................Phellinotus scaber*2′. Hymenophore brown, flat to convex.....................................................................................................................................................................33. Tubes distinctly stratified*......................................................................................................................................................Phellinotus piptadeniae*3′. Tubes indistinctly stratified.....................................................................................................................................................................................44. Pores 1–2 per mm *.............................................................................................................................................................Phellinotus magnoporatus*4′. Pores >3 per mm.......................................................................................................................................................................................................55. Pores 3–4 per mm *...............................................................................................................................................................Phellinotus xerophyticus*5′. Pores 4–6 per mm.....................................................................................................................................................................................................66. The pileal surface cracked, species distributed in the Caribbean region, North and Central America *.............................Phellinotus badius*6′. The pileal surface rimose, species distributed in South America...................................................................................................................... 77. Pileal surface black, the contex with a mycelial core and dark lines, species distributed in the Caatinga biome of northeast Brazil *....................................................................................................................................................................................................... Phellinotus neoaridus*7′. Pileal surface dark grayish brown, the contex with dark lines, species distributed in the SDTFs of South America *......................................................................................................................................................................................................... Phellinotus teixeirae*


## Figures and Tables

**Figure 1 jof-08-00216-f001:**
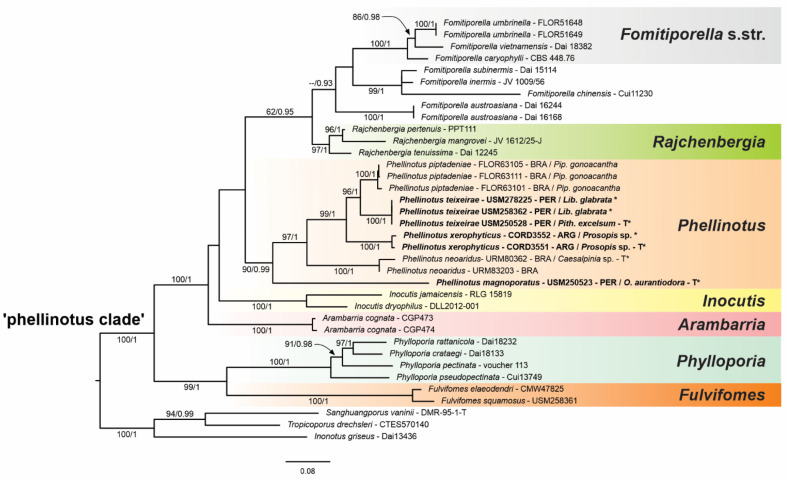
Maximum likelihood (ML) tree of the ‘phellinotus clade’ from the dataset of 36 combined sequences (*nLSU*+*nITS*+TEF1-α+RPB2). Bootstrap values above 60% and Bayesian posterior probability above 0.8 are shown. Bold font is used to indicate the newly described species. *Pip*. = *Piptadenia*, *Pith*. = *Pithecellobium*, *Lib*. = *Libidibia*, *O*. = *Ocotea*, T = material type, * = specimen of local type, BRA = Brazil, ARG = Argentina, PER = Peru.

**Figure 2 jof-08-00216-f002:**
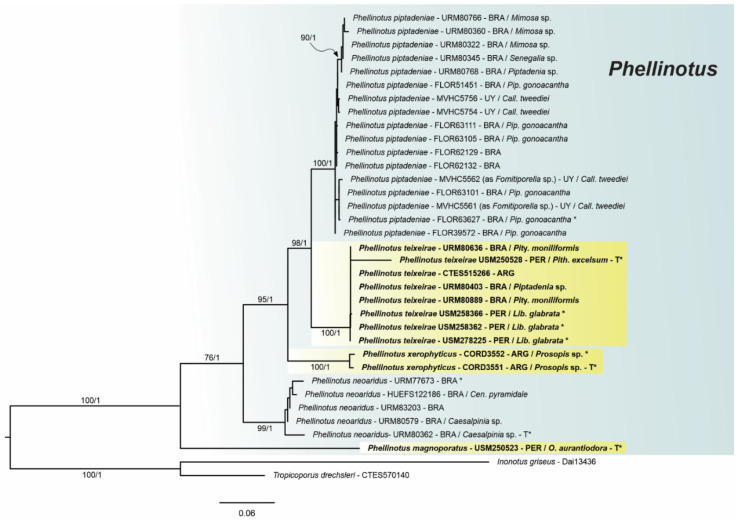
Maximum Likelihood (ML) tree of *Phellinotus* from the dataset of 35 combined sequences (*nLSU*+*nITS*+TEF1-α+RPB2). Bootstrap values above 60% and Bayesian posterior probability above 0.8 are shown. Bold font is used to indicate the newly described species. *Pip*. = *Piptadenia*, *Call*. = *Calliandra*, *Pity*. = *Pityrocarpa*, *Pith*. = *Pithecellobium*, *Lib*. = *Libidibia*, *Cen*. = *Cenostigma*, *O*. = *Ocotea*, T = material type, * = specimen of local type, BRA = Brazil, UY = Uruguay, ARG = Argentina, PER = Peru.

**Figure 3 jof-08-00216-f003:**
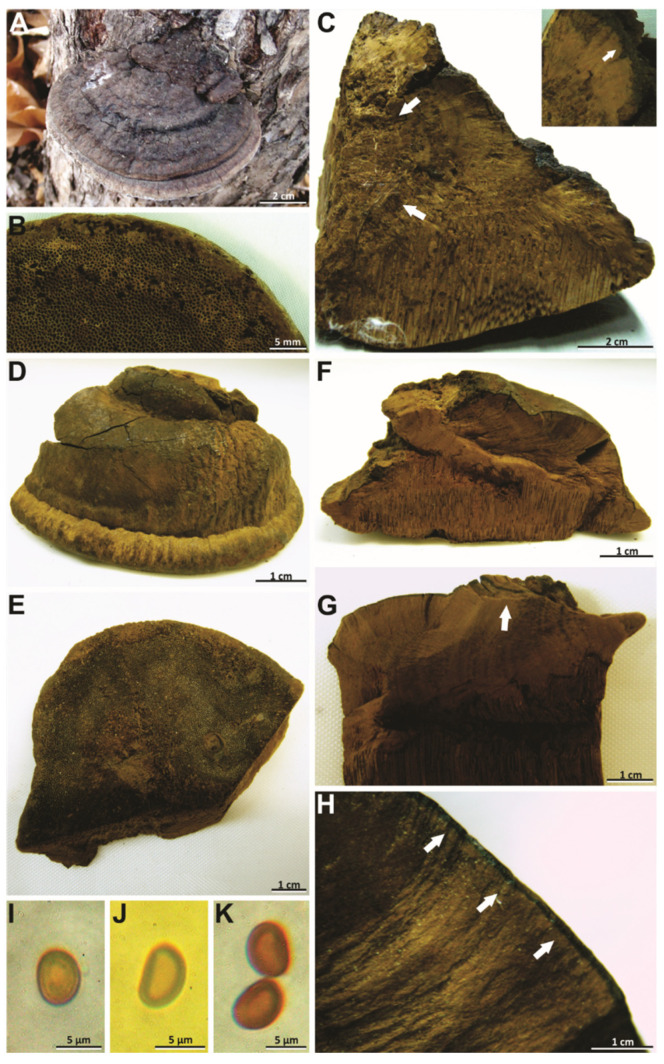
Morphological features of *Phellinotus badius* and *P*. *magnoporatus*. (**A**–**C**) *P*. *magnoporatus* (USM 250523, holotype). (**A**) basidioma; (**B**) pore surface; (**C**) perennial basidioma with detail of mycelial core in the context (the white arrows point to the cuticule and mycelial core in the context). (**D**–**K**) *Phellinotus badius* (K–M 199720, lectotype). (**D**) basidioma; (**E**) pore surface; (**F**–**H**) perennial basidioma with detail of a dark line in the context (the white arrows point to the dark line in the context); (**I**–**K**) basidiospores. (**I**) in water; (**J**) in Melzer’s reagent; (**K**) in KOH.

**Figure 4 jof-08-00216-f004:**
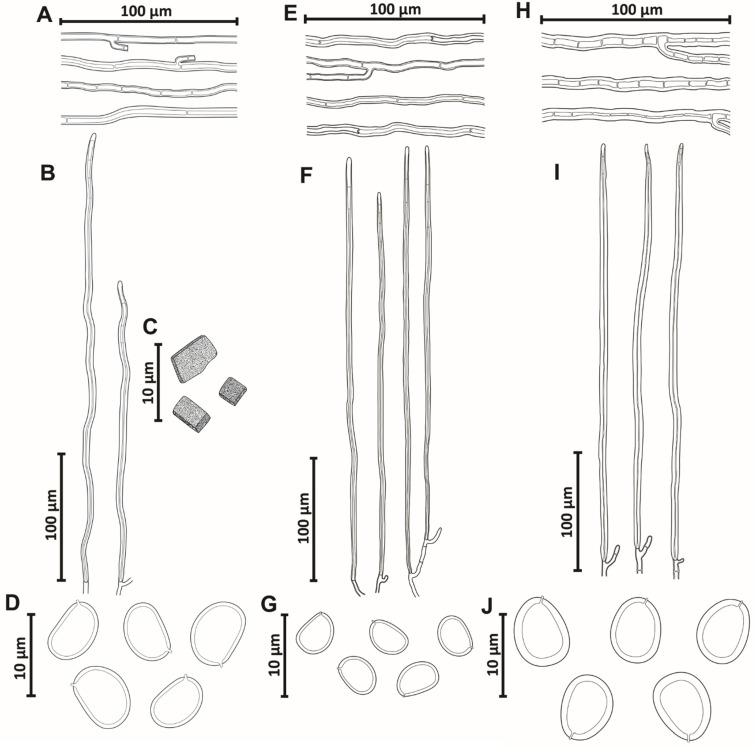
Micromorphological features of *Phellinotus badius*, *P*. *magnoporatus*, and *P*. *resinaceus*. (**A**–**D**) *Phellinotus badius*. (**A**) Generative hyphae of the context; (**B**) skeletal hyphae of the tubes; (**C**) basidiospores; (**D**) quadrangular crystals of the tubes. (**E**–**G**) *Phellinotus magnoporatus*. (**E**) Generative hyphae of the context; (**F**) skeletal hyphae of the tubes; (**G**) basidiospores. (**H**–**J**) *Phellinotus resinaceus*. (**H**) generative hyphae of the context; (**I**) skeletal hyphae of the tubes; (**J**) basidiospores.

**Figure 5 jof-08-00216-f005:**
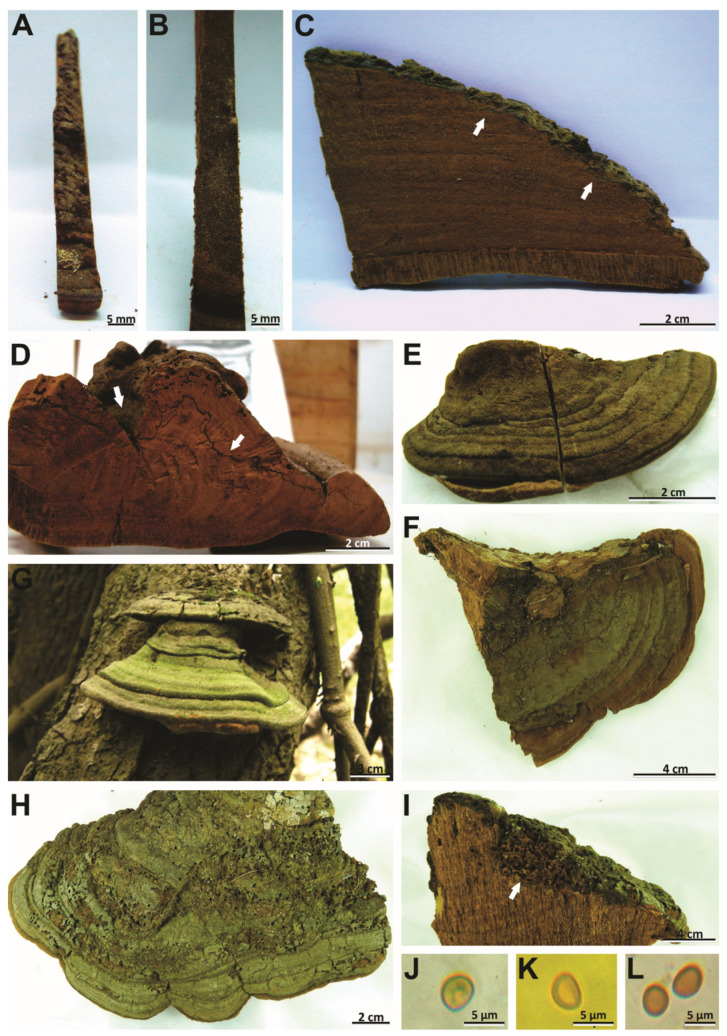
Morphological features of *Phellinotus piptadeniae*. (**A**–**C**) F 15071 (isotype). (**A**) pileal surface; (**B**) pore surface; (**C**) perennial basidioma with a dark line in the context and tubes distinctly stratified. (**D**) IAC 4365 (paratype): perennial basidioma with detail of the deep furrows on the lobulate pileal surface, a dark line in the context and tubes distinctly stratified. (**E**) URM80768: young specimen with pubescent, ondulate, slightly zoned, and brown pileal surface. (**F**) FLOR 39572: specimen with slightly cracked pileal surface. (**G**) MVH C5751: specimen with slightly lobulate pileal surface. (**H**,**I**) FLOR 39572: well-developed specimen with cracked, lobulate and olive gray pileal surface, and a dark line in the context and tubes distinctly stratified. (**J**–**L**) Basidiospores (F 15071, isotype); (**J**) in water; (**K**) in Melzer’s reagent. (**L**) in KOH.

**Figure 6 jof-08-00216-f006:**
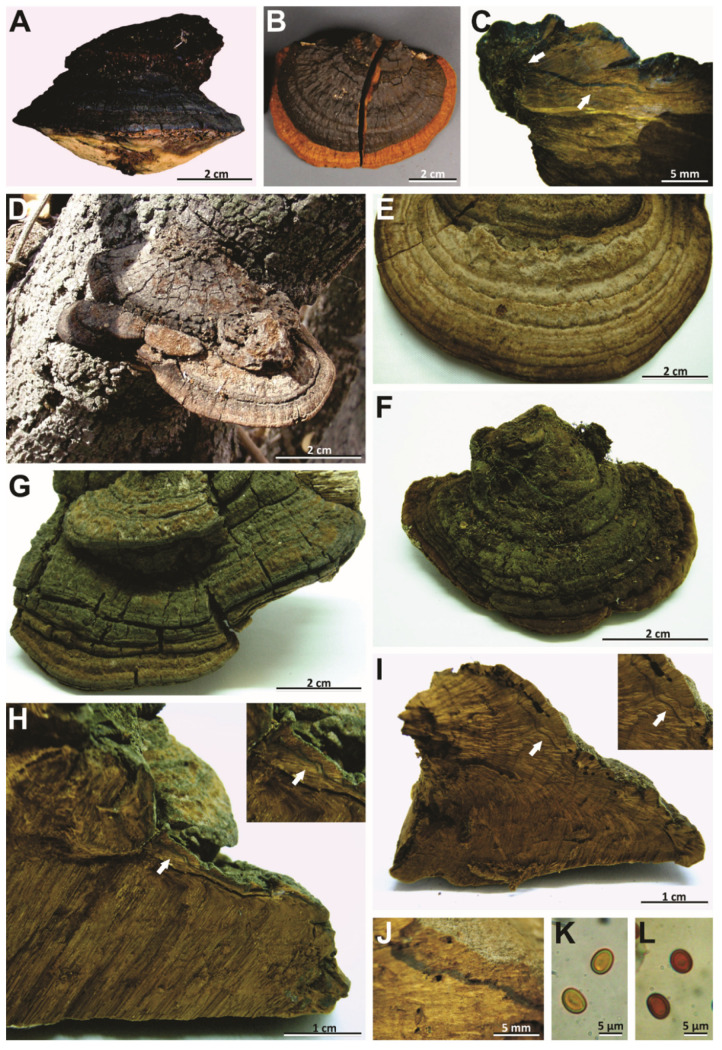
Morphological features of *Phellinotus neoaridus* and *P*. *teixeirae*. (**A**–**C**) *Phellinotus neoaridus*. (**A**) HUEFS 112186: basidioma with rimose and black pileal surface; (**B**,**C**) URM 80422. (**B**) basidioma; (**C**) the context with detail of a dark line and mycelial core. (**D**–**L**) *Phellinotus teixeirae*. (**D**–**J**) basidioma. (**D**) USM 250528 (holotype); (**E**) URM 80768: young specimen with pubescent, ondulate, slightly zoned, and brown pileal surface; (**F**) CTES 569014: specimen with slightly cracked pileal surface; (**G**) CTES 515266: well-developed specimen with rimose and grayish brown pileal surface. (**H**–**J**) Perennial basiomata with dark lines in the context and tubes indistinctly stratified. (**H**) CTES 515266; (**I**) CTES 569014; (**J**) USM 250528 (holotype). (**K**,**L**) Basidiospores. (**K**) in water; (**L**) in KOH.

**Figure 7 jof-08-00216-f007:**
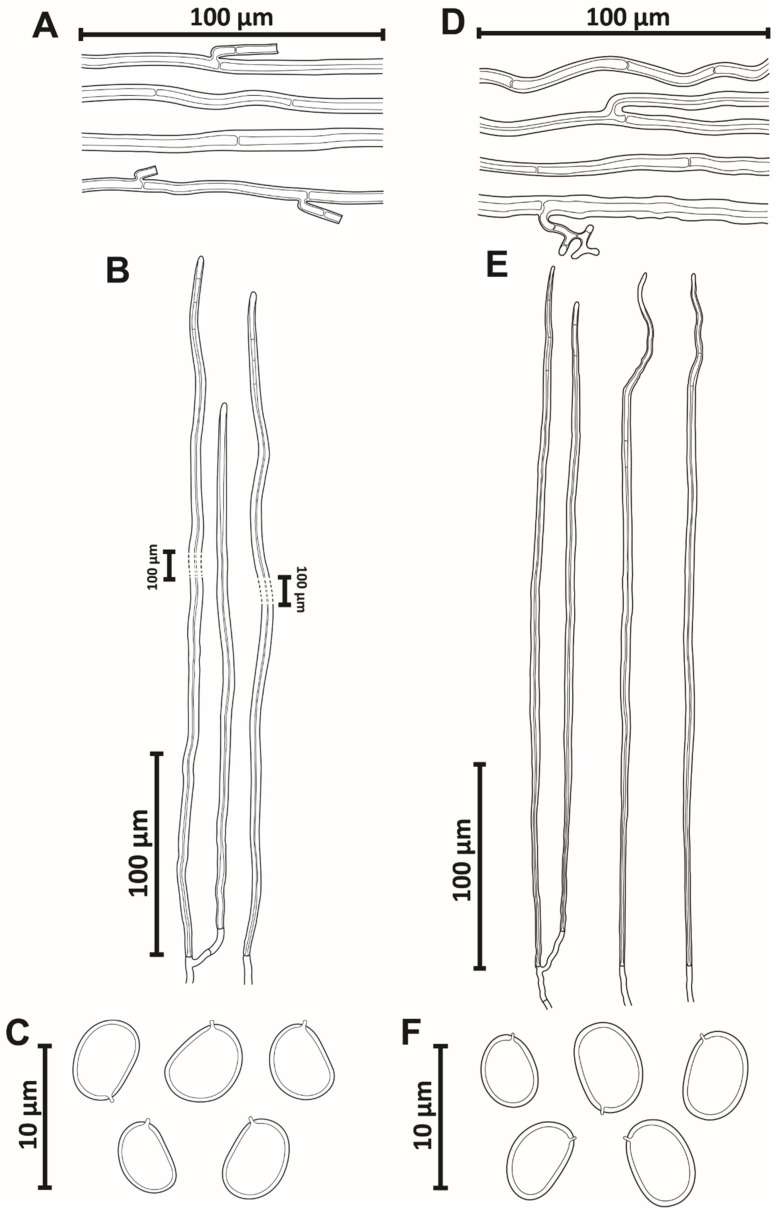
Micromorphological features of *Phellinotus piptadeniae* and *P*. *teixierae*. (**A**–**C**) *Phellinotus piptadeniae*. (**A**) generative hyphae of the context; (**B**) skeletal hyphae of the tubes; (**C**) basidiospores. (**D**–**F**) *Phellinotus teixeirae*. (**D**) Generative hyphae of the context; (**E**) skeletal hyphae of the tubes; (**F**) basidiospores.

**Figure 8 jof-08-00216-f008:**
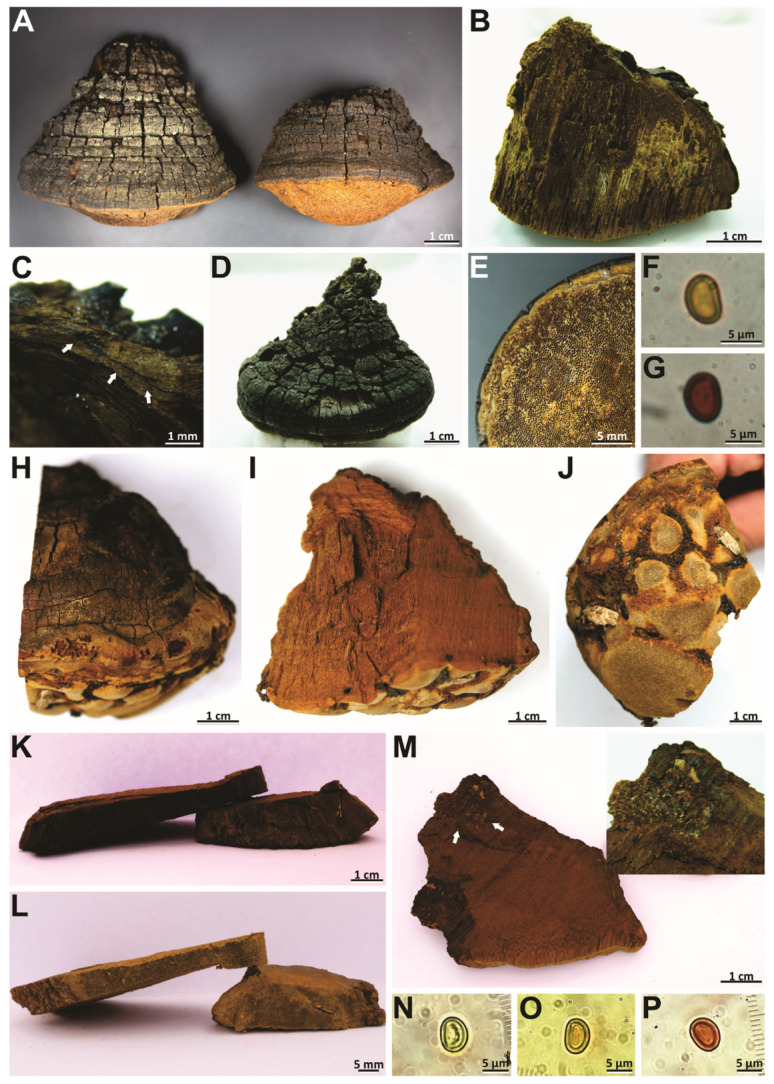
Morphological features of *Phellinotus resinaceus* and *P*. *xerophyticus*. (**A**–**G**) *Phellinotus xerophyticus*. (**A**–**C**) CORD 3552. (**A**) basidioma with rimose pileal surface; (**B**,**C**) perennial basidioma with a dark line in the context; (**D**–**G**) CORD 3551: (**D**) basidioma with rimose pileal surface; (**E**) pore surface; (**F**,**G**) basidiospores. (**F**) in water; (**G**) in KOH. (**H**–**P**) *Phellinotus resinaceus*. (**H**–**J**) K–M: 180663 (holotype); (**H**–**I**) perennial basidioma with cracked pileal surface; (**J**) pore surface; (**K**–**M**) PRM 671088 (paratype). (**K**) Cracked pileal surface; (**L**) pore surface; (**M**) perennial basidioma with detail of mycelial core in the context. (**N**–**P**) Basidiospores (**K**–**M**: 180663, holotype). (**N**) In the water; (**O**) in Melzer’s reagent, (**P**) in KOH.

**Figure 9 jof-08-00216-f009:**
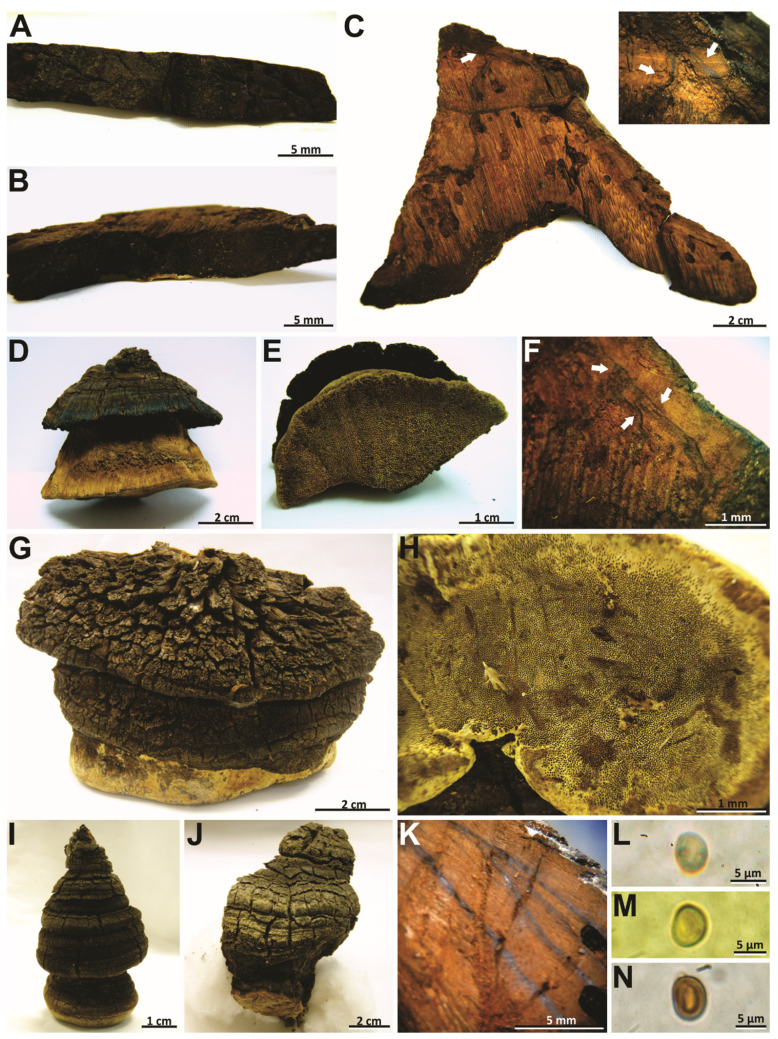
Morphological features of *Phellinotus scaber*. (**A**–**C**) K–M: 180666 (lectotype). (**A**) Pileal surface; (**B**) pore surface; (**C**) perennial basidioma with concave pore surface and dark lines in context; (**D**–**F**) MEL 229175. (**D**) Basidioma with cracked pileal surface; (**E**) concave and brown pore surface; (**F**) context with dark lines; (**G**,**H**) MEL 2359296. (**G**) Basidioma with rimose pileal surface; (**H**) concave and olive brown pore surface; (**I**) MEL 2264870; (**J**,**K**) MEL 2032383. (**J**) Basidioma; (**K**) context with dark lines; (**L**–**N**) basidiospores (**K**–**M**: 180666, lectotype). (**L**) In water; (**M**) in Melzer’s reagent; (**N**) in KOH.

**Figure 10 jof-08-00216-f010:**
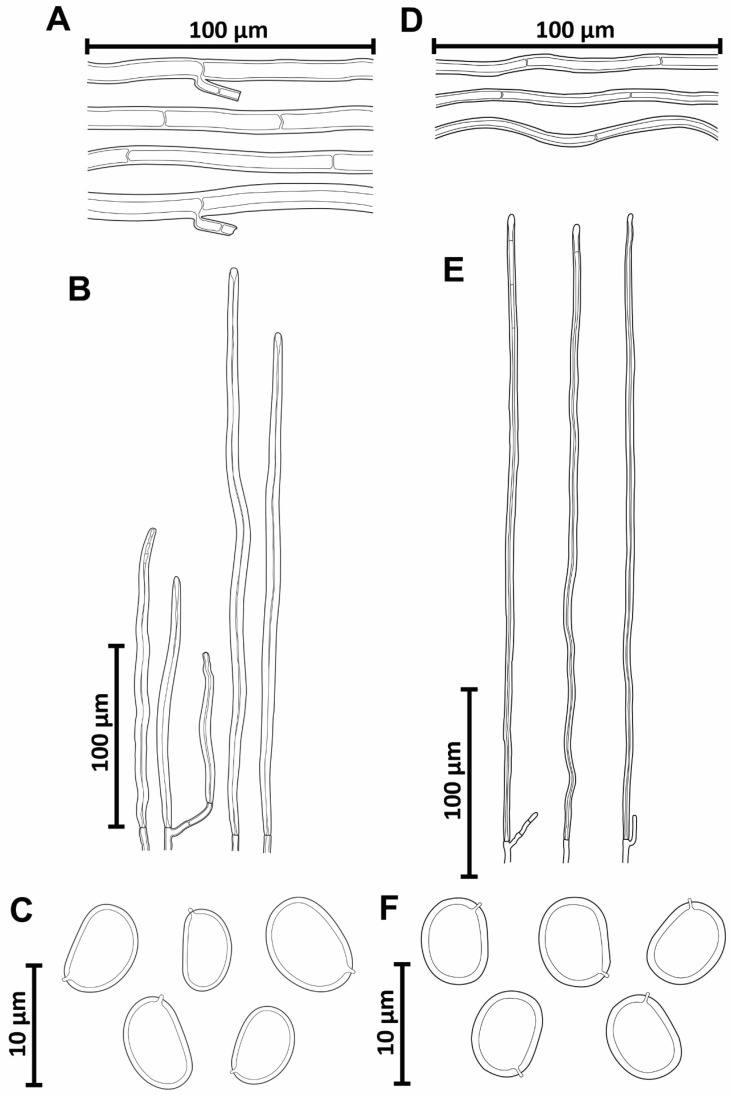
Micromorphological features of *Phellinotus scaber* and *P*. *xerophyticus*. (**A**–**C**) *Phellinotus scaber*. (**A**) Generative hyphae of the context; (**B**) skeletal hyphae of the tubes; (**C**) basidiospores. (**D**–**F**) *Phellinotus xerophyticus*. (**D**) Generative hyphae of the context; (**E**) skeletal hyphae of the tubes; (**F**) basidiospores.

**Figure 11 jof-08-00216-f011:**
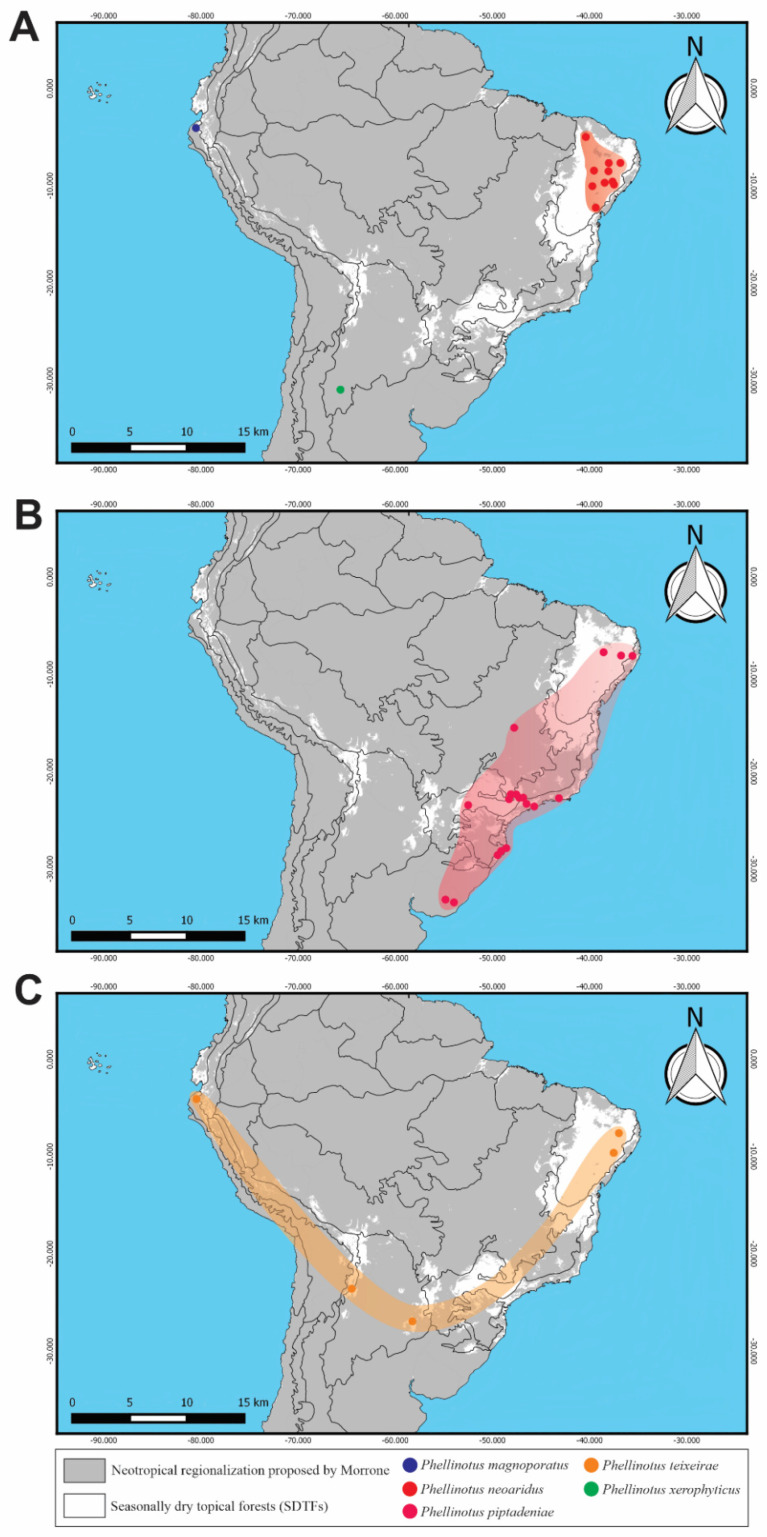
Comparison of the geographical distribution of *Phellinotus* species in South America: (**A**) *Phellinotus magnoporatus*, *P*. *neoaridus*, and *P*. *xerophyticus*; (**B**) *Phellinotus piptadeniae*; (**C**) *Phellinotus teixeirae* (the colored shadows show the distribution patterns of *Phellinotus* species).

**Table 2 jof-08-00216-t002:** Comparison of morphological features in Phellinotus species.

Species	Pileal Surface	Pore Surface		Context	Tubes	Average (µm)	Basidiospores	
		Form	Pores/mm				Size (µm)	Q Value
*P*. *badius*	C	F	4–5	Dl	Is	7.2 × 5.3	6–7(–8) × (4.5–)5–6	1.25–1.50
*P*. *magnoporatus*	Z	F	1–2	Mc	Is	5.0 × 4.0	(4–)4.5–5.5(–6) × (3–)4–4.5	1.10–1.50
*P*. *neoaridus*	R	F-Cv	(3–)4–5(–6)	Dl/Mc	Is	5.8 × 4.7	(4.5–)5.5–6.5(–7) × (3–)4–5(–6)	1.20–1.40
*P*. *piptadeniae*	C/Lb	F	(3–)4–6(–7)	Dl	Ds	5.2 × 4.0	(4–)4.5–5.5(–6) × (3–)3.5–4.5(–5)	1.10–1.50
*P*. *resinaceus*	C/Rs	F-Cv	(2–)3–4(–5)	Mc/Sr	Is	7.1 × 5.6	(6–)6.5–7.5(8) × (4.5–)5–6(–6.5)	1.17–1.44
*P*. *scaber*	R	F-Cc	3–4(–5)	Dl	Is	6.3 × 4.6	(4.5–)5.5–8(–8.5) × (3–)4–6(–6.5)	1.09–1.86
*P*. *teixeirae*	R	F	(3–)4–6(–7)	Dl	Is	5.6 × 4.2	(4–)4.5–6.5(–7) × (3–)3.5–5.5(–6)	1.10–1.57
*P*. *xerophyticus*	R	F-Cv	3–4(–5)	Dl	Is	6.8 × 5.3	(6–)6.5–7(–7.5) × (4.5–)5–5.5(–6)	1.20–1.40

C = Cracked; Z = Zonated, R = Rimose; Lb = Lobulate; Rs =Resinous substances; F = Flat; Cv = Convex; Cc = Concave; Dl = Dark line; Mc = Mycelial core; Is = Indistinctly stratified; Ds = Distinctly stratified.

## Data Availability

Publicly available datasets were analysed in this study. This data can be found here: https://www.ncbi.nlm.nih.gov/; http://www.mycobank.org/; https://www.treebase.org/treebase-web/home.html (accesed on 2 January 2022).

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
