# Peer review of "Neotropical Studies on Hymenochaetaceae: Unveiling the Diversity and Endemicity of Phellinotus"

_jof, 2022, doi:10.3390/jof8030216_

Round 1
Reviewer 1 Report
The authors have improved the manuscript, and my questions (or concerns) are answered properly.
One minior case, the doi for "reference 78" is now "https://doi.org/10.1007/s13225-021-00496-4" rather than " https://doi.org/10.21203/rs.3.rs-719853/v1"
Author Response
Response to Reviewer 1 Comments
Point 1: The authors have improved the manuscript, and my questions (or concerns) are answered properly. One minior case, the doi for "reference 78" is now "https://doi.org/10.1007/s13225-021-00496-4" rather than " https://doi.org/10.21203/rs.3.rs-719853/v1"
Response 1: We thank the reviewer 1 for their comments and suggestions to improve the manuscript. However, the “reference 78” was retired in the new version of manuscript.
Reviewer 2 Report
"Neotropical studies on Hymenochaetaceae: unveiling a diversity and endemicity in Phellinotus"
The authors report 3 new species in Phellinotus, a genus described recently (2016) by them, and provide data about Phellinotus species distribution and ecology. The paper is well written, methods used are O.K., morphological features thoroughly and professionally described, incl. figures and photos. With no sequences available, I cannot tell more about these new species but I believe they are O.K. Also, TYPE study revealed that Phellinus (Fulvifomes) badius (USA), as well as P. resinaceus (PAPUA) and P. scaber (TASMANIA) belong to their Phellinotus. The last two I do not know but P. badius transfer to Phellinotus (Fomitiporella) is surely right.
Nevertheless, the substantional part of the paper authors devoted to the ardent defence of their own problematic genus Phellinotus and there are points here that I cannot accept without criticism.
Previous molecular studies (starting with Wagner & Fischer 2002) showed 4 related Phellinus s.l. genera that all lack setae and have thickwalled, colored and ellipsoid to subglobose spores: Phylloporia, Fulvifomes, Fomitiporella and Inocutis, the species of which comprise 4 distinct phylogenetic clades and show morphological features enabling to recognize these genera easily in the field. This taxonomy is stable for years and very practical and I think that it should stay so.
Fomitiporella for example is distinct by deeply colored spores (which are also extremely abundant), fragile tubes with glancing pores, and resupinate growth with very thin subiculum. Phellinotus is in my opinion just a pileate Fomitiporella. In the original phylogeny (Ref. 1) it looks separated from Fomitiporella and more related to Inocutis just because of limited sampling, but in subsequent more complex phylogenies (Ref. 3 and this manuscript) it is distant from Inocutis and closely related to Fomitiporella, which was however split in several smaller, I think also superfluous genera (Arambaria, Rajchenbergia, ...). But monophyly in some group is in itself not sufficient for new Genus definition; it is also important what happens with related species!
The result of this splitting is that some classical Fomitiporella (e.g. F. inermis) show separated clades and dont have now their genus! Also, I am afraid that every newly collected species can destroy the monophyly of these small genera. In this respect, Phellinus sanctigeorgii (Pat.) Ryvarden seems to me especially disturbing; judging by Ryvardens descriptions it is a typical "Phellinotus" and the species is in my experience common in South-American cloud forests (but still without a sequence in GenBank). Even if distinctly pileate, in my phylogeny it is much closer to Fomitiporella s. stricto than to Phellinotus. And there is no mention about this species in the manuscript.
Also, Phellinus badius was rightly transferred to Fomitiporella (as Phellinotus) but its sequence is not shown in Phylogeny pictures (Fig. 1, 2). Perhaps as it more close to "Rajchenbergia" than "Phellinotus"? (as least in my phylogeny).
In short, I believe that splitting Fomitiporella in several genera, even if possible, brings more problems than solutions.
Also, in a short time, world-wide overwiev of Phellinus will be published in Fungal Diversity, where no splitting of Fomitiporella is recommended after extensive sampling and multigene phylogeny.
I am well avare that taxonomical rules enable everybody to express his own view on taxonomical arrangement (in appropriate form) and I cannot urge the authors to change their views. But to recommend for publication, I ask the authors:
1) to relativize a bit their decision that Phellinotus is a good genus- e.g. as a temporary concept based on available data; and to shorten a bit long discussions on this problem.
2) to discuss the Phellinus sanctigeorgii position. I am sure they have this species in their herbaria.
Author Response
Response to Reviewer 2 Comments
Point 1: The authors report 3 new species in Phellinotus, a genus described recently (2016) by them, and provide data about Phellinotus species distribution and ecology. The paper is well written, methods used are O.K., morphological features thoroughly and professionally described, incl. figures and photos. With no sequences available, I cannot tell more about these new species but I believe they are O.K. Also, TYPE study revealed that Phellinus (Fulvifomes) badius (USA), as well as P. resinaceus (PAPUA) and P. scaber (TASMANIA) belong to their Phellinotus. The last two I do not know but P. badius transfer to Phellinotus (Fomitiporella) is surely right. Nevertheless, the substantional part of the paper authors devoted to the ardent defence of their own problematic genus Phellinotus and there are points here that I cannot accept without criticism.
Response 1: We thank the reviewer 2 for the comment, and we have rewritten the main objectives of our work (please see the Introduction section) and also improved the Discussion section in the new version of manuscript (Please see the Lines 939-962).
Point 2: Previous molecular studies (starting with Wagner & Fischer 2002) showed 4 related Phellinus s.l. genera that all lack setae and have thickwalled, colored and ellipsoid to subglobose spores: Phylloporia, Fulvifomes, Fomitiporella and Inocutis, the species of which comprise 4 distinct phylogenetic clades and show morphological features enabling to recognize these genera easily in the field. This taxonomy is stable for years and very practical and I think that it should stay so.
Response 2: We understand the position of the reviewer. In this study submitted to JoF, we also accepted some of the great genera mentioned by the reviewer, such as Inocutis, Phylloporia and Fulvifomes, which are currently accepted in the taxonomy. However, we also believe that changes in the phylogenies of the large genera (i.e. Phylloporia, Fulvifomes, Inocutis and Fomitiporella) may occur as more genes and specimens of these taxa are incorporated. Previous studies, such as Zhou (2014), based on morphological and molecular data, mentions that genera with colored spores in Hymenochaetaceae (i.e. Fomitiporella, Fulvifomes, Inocutis and Phylloporia), accepted and used by current taxonomy, do not appear to be natural. Subsequently, Pildain et al. (2018) when conducting a discussion of the genus Fomitiporella, observed some inconsistencies in the monophyly of this genus. For this reason, Pildain et al. (2018) mention that considering all the genera of the 'phellinotus clade' as different taxa is the best option. In addition, Pildain et al. (2018) mention that if one admits a large Fomitiporella genus, probably we are unable to observe which biological and morphological features are leading the evolution of this group of Hymenochaetaceae and, probably, we will never recognize well the geographical processes that have conducted the diversification of this group. Therefore, despite we keep some large genera such as Fulvifomes, Inocutis and Phylloporia, we also consider the genera of 'phellinotus clade' as different taxa. This is shown in our phylogenetic inferences (in the new Figure 1) where all the genera of the 'phellinotus clade' (included Phellinotus) are strongly supported.
References: Zhou, L.W. Notes on the taxonomic positions of some Hymenochaetaceae (Basidiomycota) species with colored basidiospores. Phytotaxa 2014, 177, 183–187. https://doi.org/10.11646/ phytotaxa.177.3.7
Pildain, M.B.; Reinoso Cendoya, R.; Ortiz-Santana, B.; Becerra, J.; Rajchenberg, M. A discussion on the genus Fomitiporella (Hymenochaetaceae, Hymenochaetales) and first record of F. americana from southern South America. MycoKeys 2018, 38, 77–91. https ://doi.org/10.3897/mycok eys.38.27310
Point 3: Fomitiporella for example is distinct by deeply colored spores (which are also extremely abundant), fragile tubes with glancing pores, and resupinate growth with very thin subiculum. Phellinotus is in my opinion just a pileate Fomitiporella. In the original phylogeny (Ref. 1) it looks separated from Fomitiporella and more related to Inocutis just because of limited sampling, but in subsequent more complex phylogenies (Ref. 3 and this manuscript) it is distant from Inocutis and closely related to Fomitiporella, which was however split in several smaller, I think also superfluous genera (Arambaria, Rajchenbergia, ...). But monophyly in some group is in itself not sufficient for new Genus definition; it is also important what happens with related species!
Response 3: We thank the reviewer 2 for the comment. In the reference 1, Drechsler-Santos et al. (2016) show to Phellinotus closely related to Inocutis, as is mentioned by the reviewer. However, the reference 1 also shows the unresolved lineages of Fomitiporella s.l. proposed by Salvador-Montoya et al. (2020). For example, Fulvifomes inermis (currently as Fomitiporella inermis) grouped with specimens of Hymenochaetaceae s.p. in a supported clade (BS=100/BPP=1), and these taxa are distant from the others specimens of Fomitiporella, such as Fomitiporella sp. and Fulvifomes chinensis (currently as Fomitiporella chinensis). In addition, Fulvifomes inermis, Fulvifomes chinensis, and other taxa of Hymenochaetaceae formed a group together with species of Phellinotus and Inocutis in an unsupported clade (BS=59/BPP=0.97), i.e. Fomitiporella s.l. Then, Pildain et al. (2018) (Reference 2) show to Phellinotus closely to related Inocutis, and these genera as a sister group of the taxa of Fomitiporella and Arambarria (Fomitiporella s.l.). In addition, the taxa of Fomtiporella and Arambarria formed an unsupported clade (--/BPP=0.96), i.e. Fomitporella s.l. Therefore, Pildain et al. (2018) proposed three scenarios. Despite the morphological and phylogenetic variability, the authors decided to consider all the lineages of the ‘phellinotus clade’ as different genera as the best option. Finally, Salvador-Montoya et al. (2020) show Phellinotus distant from Inocutis (as is mentioned by the reviewer). However, the former is closely related to Rajchenbergia and distant from Fomitiporella (Fomitporella s.str.). In addition, the unresolved lineages of Fomitiporella s.l., Phellinotus, Rajchenbergia, and Arambarria formed an unsupported clade (--/BPP=0.92), i.e Fomitporella s.l. Therefore, Salvador-Montoya et al. (2020) decided to follow the first scenario proposed by Pildain et al. (2018), i.e. all lineages of the ‘phellinotus clade’ as different genera.
Based on the recommendations proposed by Pildain et al. (2018), we performed phylogenetic inferences with four-loci datasets. Also, based on the comment of the reviewer, we improved our phylogenetic inferences in the new version of the manuscript. In the new Figure 1, Phellinotus (BS=90/BPP=0.99) is related to Inocutis and distant related to unresolved lineages of Fomitiporella s.l., as well as Fomitiporella s.str. Therefore, we decided to continue with the first scenario proposed by Pildain et al. (2018) in this new version of manuscript.
References: Drechsler-Santos, E.R.; Robledo, G.L.; Lima-Júnior, N.C.; Malosso, E.; Reck, M.A.; Gibertoni, T.B.; Cavalcanti, M.A.Q.; Rajchenberg, M. Phellinotus, a new neotropical genus in the Hymenochaetaceae (Basidiomycota, Hymenochaetales). Phytotaxa, 2016, 261, 218–239. https://doi.org/10.11646/phytotaxa.261.3.2
Pildain, M.B.; Reinoso Cendoya, R.; Ortiz-Santana, B.; Becerra, J.; Rajchenberg, M. A discussion on the genus Fomitiporella (Hymenochaetaceae, Hymenochaetales) and first record of F. americana from southern South America. MycoKeys 2018, 38, 77–91. https ://doi.org/10.3897/mycok eys.38.27310
Salvador-Montoya, C.A.; Popoff, O.F.; Góes-Neto, A.; Drechsler-Santos, E.R. Global phylogenetic and morphological reassessment of Fomitiporella s.l. (Hymenochaetales, Basidiomycota): taxonomic delimitation of Fomitiporella s.s. and segregation of Rajchenbergia, gen. nov. Pl. Syst. Evol. 2020, 306, 1–27. https://doi.org/10.1007/s00606-020-01648-w
Point 4: The result of this splitting is that some classical Fomitiporella (e.g. F. inermis) show separated clades and dont have now their genus! Also, I am afraid that every newly collected species can destroy the monophyly of these small genera. In this respect, Phellinus sanctigeorgii (Pat.) Ryvarden seems to me especially disturbing; judging by Ryvardens descriptions it is a typical "Phellinotus" and the species is in my experience common in South-American cloud forests (but still without a sequence in GenBank). Even if distinctly pileate, in my phylogeny it is much closer to Fomitiporella s. stricto than to Phellinotus. And there is no mention about this species in the manuscript.
Response 4: We thank the reviewer 2 for the comment to improve the manuscript. Based on reinterpretation of literature (Lowe 1957, Ryvarden 1972, Ruiz & Valera 2006), P. sancti-georgii presents characters of the lineages of ‘phellinotus clade’. However, this species has a dimitic hyphal system (Ruiz & Valera 2006), while the species of Phellinotus have a monomitic hyphal system in the context and dimitic in the tubes. Based on Salvador-Montoya et al. (2020), the taxa in the unresolved lineages of Fomitiporella s.l. present species with a dark line in the context and a dimitic hyphal system, for example Fomitiporella chinensis. Therefore, we believe that P. sancti-georgii could be a species of the unresolved lineages of Fomitiporella s.l. (based on Salvador-Montoya et al. 2020), and it is not a species of Phellinotus.
References: Lowe, J.L. Polyporaceae of North America. The genus Fomes; Syracuse, State University College of Forestry, Syracuse University; 1957, pp. 1–97.
Ryvarden, L. A critical checklist of the Polyporaceae in tropical East Africa. Norwegian Journal of Botany 1972, 19, 229–238.
Ruiz, A. Varela, A. New reports of Aphyllophorales (Basidiomicota) in humid and cloudy montane forests from Colombia. Caldasia 2006, 28, 259–266.
Salvador-Montoya, C.A.; Popoff, O.F.; Góes-Neto, A.; Drechsler-Santos, E.R. Global phylogenetic and morphological reassessment of Fomitiporella s.l. (Hymenochaetales, Basidiomycota): taxonomic delimitation of Fomitiporella s.s. and segregation of Rajchenbergia, gen. nov. Pl. Syst. Evol. 2020, 306, 1–27. https://doi.org/10.1007/s00606-020-01648-w
Point 5: Also, Phellinus badius was rightly transferred to Fomitiporella (as Phellinotus) but its sequence is not shown in Phylogeny pictures (Fig. 1, 2). Perhaps as it more close to "Rajchenbergia" than "Phellinotus"? (as least in my phylogeny).
Response 5: We thank the reviewer 2 for the comment. Unfortunately, sequences of a P. badius specimen was not possible to obtain in this study. We revised the Genbank database (NCBI), and it has sequences of specimens previously determined as "Phellinus badius", which are LDCMY36, LDCMY31, LDCMY27, and CBS449.76. Nonetheless, LDCMY36, LDCMY31 and LDCMY27 specimens are from India (P. badius is distributed in North and Central America), and there was no information about the locality of and CBS449.76. Moreover, these specimens (i.e. LDCMY36, LDCMY31, LDCMY27 and CBS449.76) do not exhibit morphological information to check if these fit in the morphological concept of the type material of P. badius.
The reviewer 2 mentions that some specimens of P. badius were tested in the phylogenetic analyses of the reviewer, and they are closely related to Rajchenbergia than Phellinotus. When the specimens of the reviewer are published, we will compare these specimens with our data of the 'phellinotus clade' of this manuscript, and we will analyze critically with lineages of this group.
Point 6 In short, I believe that splitting Fomitiporella in several genera, even if possible, brings more problems than solutions. Also, in a short time, world-wide overwiev of Phellinus will be published in Fungal Diversity, where no splitting of Fomitiporella is recommended after extensive sampling and multigene phylogeny.
Response 6: We thank the reviewer 2 for the comment. However, we decided to continue with the first scenario proposed by Pildain et al. (2018), corroboring our phylogenetic analysis (Drechsler-Santos et al. 2016, Salvador-Montoya et al. 2020). Our decision is supported based on the response of the third paragraph of the reviewer 2. Also, when the paper mentioned by the reviewer is published we will compare the new data with our data of the 'phellinotus clade' of this manuscript, and we will analyze critically all lineages of this group.
Reference: Pildain, M.B.; Reinoso Cendoya, R.; Ortiz-Santana, B.; Becerra, J.; Rajchenberg, M. A discussion on the genus Fomitiporella (Hymenochaetaceae, Hymenochaetales) and first record of F. americana from southern South America. MycoKeys 2018, 38, 77–91. https ://doi.org/10.3897/mycok eys.38.27310
Point 7 I am well aware that taxonomical rules enable everybody to express his own view on taxonomical arrangement (in appropriate form) and I cannot urge the authors to change their views. But to recommend for publication, I ask the authors:
1) to relativize a bit their decision that Phellinotus is a good genus- e.g. as a temporary concept based on available data; and to shorten a bit long discussions on this problem.
Response 7: We thank the reviewer 2 for the comment, and we have improved it in the Discussion section of the new version of manuscript (Please see the Lines 939-962).
Point 8 2) to discuss the Phellinus sanctigeorgii position. I am sure they have this species in their herbaria.
Response 8: We thank the reviewer 2 for the comment. However, the discussion of P. sancti-georgii in the manuscript was not added. Based on reinterpretation of literature (Lowe 1957, Ryvarden 1972, Ruiz & Valera 2006), we believe that this taxon is a species of the unresolved lineages of Fomitporella s.l., and it is not a species of Phellinotus. Our decision is supported based on the response of the fourth paragraph of the reviewer 2.
References: Lowe, J.L. Polyporaceae of North America. The genus Fomes; Syracuse, State University College of Forestry, Syracuse University; 1957, pp. 1–97.
Ryvarden, L. A critical checklist of the Polyporaceae in tropical East Africa. Norwegian Journal of Botany 1972, 19, 229–238.
Ruiz, A. Varela, A. New reports of Aphyllophorales (Basidiomicota) in humid and cloudy montane forests from Colombia. Caldasia 2006, 28, 259–266.
Reviewer 3 Report
Here is the review of the paper entitled "Neotropical studies on Hymenochaetaceae: unveiling a diversity and endemicity in Phellinotus" written by Carlos A. Salvador-Montoya and co-authors.
The aim of the study is to perform a new taxonomic assessment of the genus Phellinotus, containing neotropical wood-decaying fungi mostly found on living trees of the family Fabaceae. Two- and four-loci phylogenetic analyses were performed and the genera of the ‘Phellinotus clade’ are confirmed as monophyletic groups. The study combines morphological revision and phylogenetic analyses of Phellinotus specimens, and the three new species were described as new to science: P. magnoporatus, P. teixeirae and P. xerophyticus. For the new species, detailed descriptions of micro- and macro-characters, color photographs of basidiomata in situ, and line drawings of microscopic characters are provided. A narrower species concept of P. piptadeniae is presented. Detailed morphological studies of Phellinus badius, Ph. resinaceus and Ph. scaber supported the transfer of these species to the genus Phellinotus. Dichotomous key for the identification of the species of Phellinotus is proposed.
Authors followed the newest International code of nomenclature for algae, fungi, and plants. Descriptions and molecular analysis used in the study are well done. The English language used in the manuscript needs a partial improvement. The epithet "xerophyticus" should be changed to "xerophiticus".
All other suggested corrections/additions are included in the attached review of the manuscript file (pdf).
The paper can be accepted for publication in JoF after minor revision!
Best, Reviewer

Author Response
Response to Reviewer 3 Comments
Point 1: Here is the review of the paper entitled "Neotropical studies on Hymenochaetaceae: unveiling a diversity and endemicity in Phellinotus" written by Carlos A. Salvador-Montoya and co-authors.
The aim of the study is to perform a new taxonomic assessment of the genus Phellinotus, containing neotropical wood-decaying fungi mostly found on living trees of the family Fabaceae. Two- and four-loci phylogenetic analyses were performed and the genera of the ‘Phellinotus clade’ are confirmed as monophyletic groups. The study combines morphological revision and phylogenetic analyses of Phellinotus specimens, and the three new species were described as new to science: P. magnoporatus, P. teixeirae and P. xerophyticus. For the new species, detailed descriptions of micro- and macro-characters, color photographs of basidiomata in situ, and line drawings of microscopic characters are provided. A narrower species concept of P. piptadeniae is presented. Detailed morphological studies of Phellinus badius, Ph. resinaceus and Ph. scaber supported the transfer of these species to the genus Phellinotus. Dichotomous key for the identification of the species of Phellinotus is proposed.
Authors followed the newest International code of nomenclature for algae, fungi, and plants. Descriptions and molecular analysis used in the study are well done. The English language used in the manuscript needs a partial improvement. The epithet "xerophyticus" should be changed to "xerophiticus"
Response 1: We agreed with Reviewer 3. It was made in this new version of the manuscript (Please see the Line 780).
Point 2: All other suggested corrections/additions are included in the attached review of the manuscript file (pdf).
Response 2: We agreed with Reviewer 3. It was made in this new version of the manuscript. Below is a list of all the comments by the reviewer made on the pdf:
- with -> containing
Response: We agreed with Reviewer 3. It was made in this new version of the manuscript (Please see the Line 27 in the Abstract).
- The other -> On the other
Response: We agreed with Reviewer 3. It was made in this new version of the manuscript (Please see the Line 29 in the Abstract).
- analaysis -> analysis
Response: We agreed with Reviewer 3. It was made in this new version of the manuscript (Please see the Line 30 in the Abstract).
- P. piptadeniae presents a narrower concept in its morphology, as well in its distribution -> for P. piptadeniae a narrower species concept was adopted with redefined morphological characters and more limited distribution range,
Response: We agreed with Reviewer 3. It was made in this new version of the manuscript (Please see the Lines 36 and 37 in the Abstract).
- Suggestion: put "3 new taxa" as a first keyword
Response: We agreed with Reviewer 3. It was made in this new version of the manuscript (Please see the Line 41 in the Keywords).
- Derchsler -> Drechsler
Response: We agreed with Reviewer 3. It was made in this new version of the manuscript (Please see the Line 44).
- delete comma
Response: We agreed with Reviewer 3. It was made in this new version of the manuscript (Please see the Line 47).
- This genus emerges from a morphological and phylogenetic approach -> The genus is defined using a morphological and phylogenetic approach.
Response: We agreed with Reviewer 3. It was made in this new version of the manuscript (Please see the Line 49).
- and is related -> It is related
Response: We agreed with Reviewer 3. It was made in this new version of the manuscript (Please see the Line 49).
- s. -> s.str.
Response: We agreed with Reviewer 3. It was made in this new version of the manuscript (Please see the Lines 50, 57, 208, 220, 462, 883, 884, 897, 912, 915 and 957).
- monophyletics -> monophyletic
Response: We agreed with Reviewer 3. It was made in this new version of the manuscript (Please see the Line 55).
- interpretations -> analyses
Response: We agreed with Reviewer 3. It was made in this new version of the manuscript (Please see the Line 60).
- phylogenetic inferences and taxonomic inferences -> phylogenetic and taxonomic relationships
Response: We agreed with Reviewer 3. It was made in this new version of the manuscript (Please see the Lines 62 and 63).
- presents -> possesses
Response: We agreed with Reviewer 3. It was made in this new version of the manuscript (Please see the Lines 66 and 456).
- there is cryptic diversity within the species concept -> the current species concept does not recognize cryptic diversity in this species complex.
Response: We agreed with Reviewer 3. It was made in this new version of the manuscript (Please see the Lines 83 and 84).
- Comment: For Phellinus you may use abbreviation , which is different from P. for Phellinotus
Response: We agreed with Reviewer 3. It was made in this new version of the manuscript (Please see the Lines 91 and 92).
- Comment: please go through the text and make needed change (pilear -> pileal)
Response: We agreed with Reviewer 3. It was made in this new version of the manuscript (Please see the Lines 99, 296, 301, 320, 328, 329, 338, 348, 373, 375, 376, 377, 378, 388, 444, 434, 436, 437, 450, 457, 459, 463, 542, 556, 557, 558, 559, 598, 612, 614, 615, 642, 646, 677, 685, 712, 717, 722, 778, 785, 861, 862, 885, 898, 916, 933, 835 and Table 2).
- millimeter -> mm
Response: We agreed with Reviewer 3. It was made in this new version of the manuscript (Please see the Line 100).
- observation was from freehand sections. -> was based on freehand sections
Response: We agreed with Reviewer 3. It was made in this new version of the manuscript (Please see the Line 102).
- Question: in both directions or in one?
Response: The sequencing was performed in both directions.
- accessions -> the accession number
Response: We agreed with Reviewer 3. It was made in this new version of the manuscript (Please see the Line 201).
- entities -> taxa
Response: We agreed with Reviewer 3. It was made in this new version of the manuscript (Please see the Lines 240, 248 and 883).
- Habit -> Habitat
Response: We agreed with Reviewer 3. It was made in this new version of the manuscript (Please see the Lines 315, 369, 424, 573, 636 and 707).
- ellipsoid with a flattened side basidiospores (7–8 × 5–6 μm) -> ellipsoid basidiospores (7–8 × 5–6 μm) with flattened side
Response: We agreed with Reviewer 3. It was made in this new version of the manuscript (Please see the Line 323).
- Please use full generic name -> O.
Response: We agreed with Reviewer 3. It was made in this new version of the manuscript (Please see the Line 370).
- 4–5 -> hymenophore with 4–5
Response: We agreed with Reviewer 3. It was made in this new version of the manuscript (Please see the Line 384-385).
- of -> on
Response: We agreed with Reviewer 3. It was made in this new version of the manuscript (Please see the Lines 386, 466, 654, 813 and 870).
- Please use full generic name -> P.
Response: We agreed with Reviewer 3. It was made in this new version of the manuscript (Please see the Line 425).
- specimens - > basidiomata
Response: We agreed with Reviewer 3. It was made in this new version of the manuscript (Please see the Line 457).
- paratipo -> paratype
Response: We agreed with Reviewer 3. It was made in this new version of the manuscript (Please see the Line 469).
- and - > which
Response: We agreed with Reviewer 3. It was made in this new version of the manuscript (Please see the Line 577).
- has larger -> has hymenophore with larger
Response: We agreed with Reviewer 3. It was made in this new version of the manuscript (Please see the Line 582).
- the genus [1]. -> the genus [1] as well.
Response: We agreed with Reviewer 3. It was made in this new version of the manuscript (Please see the Line 651).
- basidiospres -> basidiospores
Response: We agreed with Reviewer 3. It was made in this new version of the manuscript (Please see the Line 726).
- Important notice. The epithet should be correctly spelled as xerophyticus not xerophiticus. You can search through Index fungorum / Mycobank. So, please change the epithet spelling throughout the text and in phylogenetic trees.
Response: We agreed with Reviewer 3. It was made in this new version of the manuscript (Please see the Lines 36, 241, 247, 555, 621, 772, 783, 808, 814, 895, 930, 932, 935, Figures 1 and 2, Tables 1 and 2).
- with 3–4 pores -> with hymenophore having 3–4 pores
Response: We agreed with Reviewer 3. It was made in this new version of the manuscript (Please see the Line 809).
- has slightly -> has hymenophore with slightly
Response: We agreed with Reviewer 3. It was made in this new version of the manuscript (Please see the Line 811).
- has slightly smaller -> has hymenophore with slightly smaller
Response: We agreed with Reviewer 3. It was made in this new version of the manuscript (Please see the Line 816).
- and basdiospores -> and has slightly smaller basidiospores
Response: We agreed with Reviewer 3. It was made in this new version of the manuscript (Please see the Line 817).
- Specimen - > Additional specimen
Response: We agreed with Reviewer 3. It was made in this new version of the manuscript (Please see the Line 822).
- variables ? maybe variability is better?
Response: We agreed with Reviewer 3. It was made in this new version of the manuscript (Please see the Line 851).
- Taxnomic -> Taxonomic
Response: We agreed with Reviewer 3. It was made in this new version of the manuscript (Please see the Line 827).
- analyzes -> analyses
Response: We agreed with Reviewer 3. It was made in this new version of the manuscript (Please see the Line 843).
- have -> contain
Response: We agreed with Reviewer 3. It was made in this new version of the manuscript (Please see the Line 846).
- narrower concept in morphology -> narrower morphological concept
Response: We agreed with Reviewer 3. It was made in this new version of the manuscript (Please see the Line 852).
- distribution - > redefined more limited distribution range
Response: We agreed with Reviewer 3. It was made in this new version of the manuscript (Please see the Line 853).
- on interpretation -> on the interpretation
Response: We agreed with Reviewer 3. It was made in this new version of the manuscript (Please see the Line 864).
- and distributed - > species distributed
Response: We agreed with Reviewer 3. It was made in this new version of the manuscript (Please see the Lines 989-994).
- analyzed -> analysed
Response: We agreed with Reviewer 3. It was made in this new version of the manuscript (Please see the Line 1014).
The paper can be accepted for publication in JoF after minor revision!
Reviewer 4 Report
MS review:
Neotropical studies on Hymenochaetaceae: unveiling a diversity and endemicity in Phellinotus
Salvador-Montoya C.A., Elias S.G., Popoff O.F., Robledo G.L., Urcelay C., Góes-Neto A., Martínez S., Drechsler-Santos E.R.
The manuscript studies a neotropical genus of wood-destroying fungi, Phellinotus, which is commonly found on living members of the Fabaceae family. Two species were originally described: P. neoaridus and P. piptadeniae. The paper presents the current ideas about the phylogeny of the genus Phellinotus. On the basis of a two-locus phylogenetic analysis, relationships with species from the genus Fomitiporella s.l. A two- and four-locus phylogenetic analysis was carried out, which made it possible to identify some monophyletic groups. Molecular analysis allowed the description of three new species in the genus Phellinotus, which are proposed to be named as P. magnoporatus, P. teixeirae and P. xerophit-icus. In addition, P. piptadeniae is a narrower concept in its morphology as well as its distribution, and both P. neoaridus and P. teixeirae have a distribution area limited to the seasonally dry tropical forests of South America. In addition, based on detailed morphological revisions, Phellinotus badius, Phellinus resinaceus, and Phellinus scaber were assigned to Phellinotus.
The authors discuss in detail the geographical distribution and range of the hosts of the genus.
Undoubtedly, the manuscript is interesting, timely and worthy of publication in ”Journal of Fungi”.
This is an extremely interesting genus of Polypores, with which there have been many problems for a long time.
Of great interest is the key-determinant for the new scope of the genus Phellinotus.
Author Response
Response to Reviewer 4 Comments
Point 1: The manuscript studies a neotropical genus of wood-destroying fungi, Phellinotus, which is commonly found on living members of the Fabaceae family. Two species were originally described: P. neoaridus and P. piptadeniae. The paper presents the current ideas about the phylogeny of the genus Phellinotus. On the basis of a two-locus phylogenetic analysis, relationships with species from the genus Fomitiporella s.l. A two- and four-locus phylogenetic analysis was carried out, which made it possible to identify some monophyletic groups. Molecular analysis allowed the description of three new species in the genus Phellinotus, which are proposed to be named as P. magnoporatus, P. teixeirae and P. xerophit-icus. In addition, P. piptadeniae is a narrower concept in its morphology as well as its distribution, and both P. neoaridus and P. teixeirae have a distribution area limited to the seasonally dry tropical forests of South America. In addition, based on detailed morphological revisions, Phellinotus badius, Phellinus resinaceus, and Phellinus scaber were assigned to Phellinotus.
The authors discuss in detail the geographical distribution and range of the hosts of the genus.
Undoubtedly, the manuscript is interesting, timely and worthy of publication in ”Journal of Fungi”.
This is an extremely interesting genus of Polypores, with which there have been many problems for a long time.
Of great interest is the key-determinant for the new scope of the genus Phellinotus.
Response 1: We thank the reviewer 4 for their comments and suggestions to improve the manuscript.
Round 2
Reviewer 2 Report
The Authors responses to my comments and objections are careful and exhaustive, but corrections in the text are minimal and I feel that my objections were rather repelled by selected arguments than seriously taken into account. I still think that the repeatedly recommended option to take all lineages as genera is a direct way to hell - I mean impractical, and taxonomy was developed primarilly for practical reasons.
Nevertheless, as this is more a matter of taxonomical ideology than of results and their interpretation, and because the paper provides many other important results and data, I do recommentd it for publication.
This manuscript is a resubmission of an earlier submission. The following is a list of the peer review reports and author responses from that submission.
Round 1
Reviewer 1 Report
This manuscript focuses on a small genus with two known species in a regional area. Three new species are described with proposals of three new combinations. The interest of this manuscript is thus limited. In addition, this manuscript has some scientific flaws as below.
THREE NEW COMBINATIONS.
The authors make combinations as previously papers: following a morphological examination. However, this method only cannot make their taxonomic position stable. I will suggest the authors propose epitypes from sequenced specimens, which are morphological identify to the current types and originates from type locality. Then, the taxonomic position of these three species will be solved once for all.
PHELLINOTUS CLADE.
This clade was proposed by previous papers and confirmed by the current study, but each genus of this clade is represented by rare species. This kind of phylogeny cannot reflect the real relationships among these genera and other related genera. For example, in the reference [54], more species were sampled from these genera and the Phellinotus clade was failed to be resolved. PS: the reference [54] is still under peer review and thus should not be cited at all, because the final version of that paper may be quite different from the current one.
PHELLINOTUS.
Within the so-called Phellinotus clade, the monophyly of Phellinotus is not strongly supported, especially in Figure 2 (we can say not supported at all). In addition, Sanghuangporus, Phylloporia and Fomitiporella are also not well supported, even if each genus is represented by VERY rare species. In one word, the taxonomic position of the three new species at the generic level is not undoubtful.
HOST & GEOGRAPHIC DISTRIBUTION.
This information is only descriptive. No analysis in methodology, like biogeography, evolution and dispersal, is utilized. Not much informative.
GENERALLY.
According to the current data, this manuscript contributes nothing deserved to be published in JoF.
Author Response
Point 1: Comments and Suggestions for Authors
This manuscript focuses on a small genus with two known species in a regional area. Three new species are described with proposals of three new combinations. The interest of this manuscript is thus limited. In addition, this manuscript has some scientific flaws as below.
Response 1: This work was submitted to Special Issue "Dimensions of Tropical Fungal Diversity" of JoF. Based on focus of Special Issue, this work fits in its requirements.
Point 2: THREE NEW COMBINATIONS.
The authors make combinations as previously papers: following a morphological examination. However, this method only cannot make their taxonomic position stable. I will suggest the authors propose epitypes from sequenced specimens, which are morphological identify to the current types and originates from type locality. Then, the taxonomic position of these three species will be solved once for all.
Response 2: We understand the position of the reviewer 1. In this work was not possible obtain sequences of specimen that fill morphological concept of type materials of P. badius, P. scaber and P. rasinaceus. However, this does not preclude the publication of these species. An important part of this work is that these species should not be considered in Phellinus (in its wide morphological concept, Phellinus s.l.). In addition, we recommended in the manuscript that inferences phylogenetic with sequences of these species will help to understand much better its taxonomic position in Hymenochaetaceae.
Point 3: PHELLINOTUS CLADE.
This clade was proposed by previous papers and confirmed by the current study, but each genus of this clade is represented by rare species. This kind of phylogeny cannot reflect the real relationships among these genera and other related genera. For example, in the reference [54], more species were sampled from these genera and the Phellinotus clade was failed to be resolved. PS: the reference [54] is still under peer review and thus should not be cited at all, because the final version of that paper may be quite different from the current one.
Response 1: We do not agree with the comment of reviewer 1. The reviewer mentions that the phylogenetic inferences to the "Phellinotus clade" in this work were made with rare species. However, these "rare" species also were used in previous works as Pildain et al. (2018), Ji et al (2017, 2018), Salvador-Montoya et al. (2020), as well as the recently work Wu et al. (under review). If we consider the comment of Reviewer 1, all these work also not reflect the real relationships among genera of Hymenochaetaceae. The reviewer mention that the reference Wu et al. (under review) is still under peer review and should not be cited. However, following Instructions of Authors in JoF, this reference can consider in the category "Unpublished materials intended for publication". Reference Wu et al. (under review) is available at: DOI: 10.21203/rs.3.rs-719853/v1. ​Finally, the "Phellinotus clade" has moderate support (BPP = 0.94) in the reference Wu et al. (under review).
Point 4: PHELLINOTUS.
Within the so-called Phellinotus clade, the monophyly of Phellinotus is not strongly supported, especially in Figure 2 (we can say not supported at all). In addition, Sanghuangporus, Phylloporia and Fomitiporella are also not well supported, even if each genus is represented by VERY rare species. In one word, the taxonomic position of the three new species at the generic level is not undoubtful.
Response 4: We are not according with the comment of the Reviewer 1. The new Figure 2 in the manuscript shown the genus Phellinotus well supported. In addition, the morphological data of species of Phellinotus support the concept and monophyly of the genus. The new Figure 2 shown to Phylloporia with strongly support (BS=100/BPP=1), and Sanghuangporus moderate supported (BPP=0.99). The latter no should be a reason to not consider the results of this work. In the case of the Fomitiporella s.l., we used only one specimen that has the four genes (i.e ITS, LSU, EF-1 and RPB2), Fomitiporella chinensis (Cui 11230). Genbank database only has this specimen of Fomitiporella s.l. with ITS, LSU, EF-1 and RPB2 regions. Finally, in the phylogenetic inferences performed in this work we used species that they were used in pervious works as Pildain et al. (2018), Ji et al (2017, 2018), Salvador-Montoya et al. (2020), and not were categorized as very rare species. This no should be as other reason to not consider the results of phylogenetic inferences of this work.
Point 5: HOST & GEOGRAPHIC DISTRIBUTION.
This information is only descriptive. No analysis in methodology, like biogeography, evolution and dispersal, is utilized. Not much informative.
Response 5: Ecological information (e.x. hosts and geographical distribution) in this work is relevant. These help to complement the taxonomic concept of species presented in this work.
Point 6: GENERALLY.
According to the current data, this manuscript contributes nothing deserved to be published in JoF.
Response 6: We are not according with the comment of the Reviewer 1. This work reveals a hidden diversity (with new lineages/entities) in Phellinotus for the tropics based on morphological, molecular and ecological data. Therefore, it contributes significantly to Special Issue of the JoF
Reviewer 2 Report
This manuscript deals with species diversity and endemicity in Phellinotus (Hymenochaetaceae) in South America. Based on morphology, ecology and molecular analyses, three species, P. magnoporatus, P. teixeirae and P. xerophiticus, are described. These results are novel and worth to be published.
Some special comments:
- I suggest author add references for each of species data in Table 1.
- Table 1 change “Tsuga Canadensis” as “Tsuga canadensis”.
- Line 276: “(150–)170–479(–595) μm long × (3.5–)4–5(–5.5) μm diam”, how did you measure the length of skeletal hyphae? The same question for other species in the following text.
- In Table 2, I don’t think the color of pore surface is important character, because they are more or less brownish, it is better to delete it.
- Line 811, how do you conclude “Phellinotus badius, P. resinaceus and P. scaber are herein presented in a correct taxonomic status without DNA data? Which one is wrong?
- Line 929 and 930, what are the differences between “pilear surface cracked” and “pilear surface rimose”?
Author Response
Point 1: I suggest author add references for each of species data in Table 1.
Response 1: It was added in the new version of the manuscript.
Point 2: Table 1 change “Tsuga Canadensis” as “Tsuga canadensis”.
Response 2: It was made in the new version of the manuscript.
Point 3: Line 276: “(150–)170–479(–595) μm long × (3.5–)4–5(–5.5) μm diam”, how did you measure the length of skeletal hyphae? The same question for other species in the following text.
Response 3: The skeletal hyphae were measured from the starting part that is close to the generative hypha to the apical part of them. This was possible after the context / tube portions passed through a 5% NaOH solution at 60 °C, for 20 min. Then, these were dissected to be observed on microscope.
Point 4: In Table 2, I don’t think the color of pore surface is important character, because they are more or less brownish, it is better to delete it.
Response 4: It was made in the new version of the manuscript.
Point 5: Line 811, how do you conclude “Phellinotus badius, P. resinaceus and P. scaber are herein presented in a correct taxonomic status without DNA data? Which one is wrong?
Response 5: We concluded, based on morphological revision from type specimens, that these species belong to Phellinotus by have a dark line and mycelial core, as well as monomitic hyphal system, in the context, dimitic hyphal system in the tubes, and yellowish basidiospores with a flattened side that turn darker in KOH solution. We understand the question of the Reviewer 2 about DNA sequences of these species and their correct taxonomic position, but also it is important that these species should not be considered in the wide concept of Phellinus (Phellinus s.l.), and “Yes” in other genera that fit in its morphological concept. In this work, we recommend that phylogenetic inferences with sequences of specimens of these species that fit in morphological concept of their type specimens could help to understand much better their taxonomic position.
Point 6: Line 929 and 930, what are the differences between “pilear surface cracked” and “pilear surface rimose”?
Response 6: According to observations of Kotlaba & Pouzar (1978) and Salvador-Montoya (2018), cracked pilear surface shows lines with slightly deeply sulci from having split in well-develop specimens. In the case of rimose pilear surface, it shows numerous lines with very deep sulci, becoming the pilear surface torn apart.
Reviewer 3 Report
This is an interesting manuscript about the diversity and distribution of Phellinotus species in South America using morphological, molecular, and ecological approaches. The manuscript brings important contributions for the knowledge of Phellinotus to the world. The descriptions are presented well and I have no doubts about the novelty of this work. Having said that, I think that a lot of aspects should be improved on to make this a very good publication. Multiple comments were made on the manuscript itself, while I highlight some concerns below.
- The main results concerning multi-loci phylogenetic analyses should also be included in the abstract.
- Keywords should be arranged in alphabetical order and the first letter of all the keywords should start with a capital letter to maintain uniformity.
- The introduction section is short. Some more information can be added to it.
- The last paragraph of the introduction section should describe the objectives of this work and not the results obtained. Make the corrections accordingly.
- “Specimens of Phellinotus were collected in different ecosystems from Argentina, Brazil, Peru and Uruguay.” The time frame needs to be added to the statement for better clarity.
- Did you try to isolate these Phellinotus species into the pure culture? Further, don’t you think that the dried pieces of basidiomata used for genomic DNA isolation can contain contamination which can influence the overall sequencing results?
- Check for xanthocroic reaction. I think it’s
- Line 98, replace extracted with extract.
- Replace ‘Bayes 3.2.6’ with ‘MrBayes 3.2.6’.
- Line 213, replace ‘datailed’ with detailed.
- Please mention the PCR mix used and PCR conditions for nITS, nLSU, TEF1-α and RPB2.
- Did you check for the purity of the obtained sequences?
- Line 151, check for ‘A total of ×107 generations’.
- In table 1, the newly added Phellinotus species from South America should be provided in bold to mark its clear distinction from others. Further, I can see that for most of the Phellinotus species GenBank accession numbers are missing. Authors should rectify these issues.
- In line 164, ‘A total of 52 new sequences, i.e., 18 nLSU, 21 nITS, nine TEF1-α and four RPB2 were generated.’ What was the total no. of samples subjected for sequencing analyses? How many strains of a particular species were subjected for analyses? Why only nine and four sequences were generated in the case of TEF1-α and RPB2, respectively. I think if you tried with some other primer pairs a better no. of sequences would be derived.
- From Table 2 legend, remove italics from ‘species’.
- In notes for Phellinotus piptadeniae, no comparison, as well as distinction with other allied species Phellinotus, was provided. I think the same will provide better clarity to the readers.
- Check for IKI- throughout the text.
- Line 579, replace ‘this taxon’ with ‘This taxon”.
- Check for gapping error on page 32.
- Line 807, check for indepented taxa.
- Please add a paragraph of conclusions based on the novelty of this kind of research and the importance of taxonomic studies to re-classify these species.
- Check for other typo and grammatical errors throughout the text.
Author Response
Response to Reviewer 3 Comments
Point 1: The main results concerning multi-loci phylogenetic analyses should also be included in the abstract.
Response 1: It was made in the new version of the manuscript.
Point 2: Keywords should be arranged in alphabetical order and the first letter of all the keywords should start with a capital letter to maintain uniformity.
Response2: It was made in the new version of the manuscript.
Point 3: The introduction section is short. Some more information can be added to it.
Response 3: It was made in the new version of the manuscript.
Point 4: The last paragraph of the introduction section should describe the objectives of this work and not the results obtained. Make the corrections accordingly.
Response4: It was made in the new version of the manuscript.
Point 5: “Specimens of Phellinotus were collected in different ecosystems from Argentina, Brazil, Peru and Uruguay.” The time frame needs to be added to the statement for better clarity.
Response 5: It was made in the new version of the manuscript.
Point 6: Did you try to isolate these Phellinotus species into the pure culture? Further, don’t you think that the dried pieces of basidiomata used for genomic DNA isolation can contain contamination which can influence the overall sequencing results?
Response 6: The consensus sequences from this work were blasted for comparison with existing sequences from Phellinotus species (P. neoaridus and P. piptadeniae) in Genbank database. The sequences that showed a coverage greater than 95% with sequences from P. neoaridus and P. piptadeniae were used in this work. Unfortunately, the pure culture of the new Phellinotus species were not obtained.
Point 7: Check for xanthocroic reaction. I think it’s
Response 7: It was made in the new version of the manuscript.
Point 8: Line 98, replace extracted with extract.
Response 8: It was made in the new version of the manuscript.
Point 9: Replace ‘Bayes 3.2.6’ with ‘MrBayes 3.2.6’.
Response 9: It was made in the new version of the manuscript.
Point 10: Line 213, replace ‘datailed’ with detailed.
Response 10: It was made in the new version of the manuscript.
Point 11: Please mention the PCR mix used and PCR conditions for nITS, nLSU, TEF1-α and RPB2.
Response 11: It was added in the new version of the manuscript.
Point 12: Did you check for the purity of the obtained sequences?
Response 12: Yes, we did. The sequences were verficate and edited in Geneious program, and consensus sequences greater than or equal to 95% high quality (HQ) were used.
Point 13: Line 151, check for ‘A total of ×107 generations’.
Response 13: It was made in the new version of the manuscript.
Point 14: In table 1, the newly added Phellinotus species from South America should be provided in bold to mark its clear distinction from others. Further, I can see that for most of the Phellinotus species GenBank accession numbers are missing. Authors should rectify these issues.
Response 14: It was made in the new version of the manuscript.
Point 15: In line 164, ‘A total of 52 new sequences, i.e., 18 nLSU, 21 nITS, nine TEF1-α and four RPB2 were generated.’ What was the total no. of samples subjected for sequencing analyses? How many strains of a particular species were subjected for analyses? Why only nine and four sequences were generated in the case of TEF1-α and RPB2, respectively. I think if you tried with some other primer pairs a better no. of sequences would be derived.
Response 15: A total of 33 samples (dried basidiomata) of species of Phellinotus were subjected for sequencing analyses, which 14 were reference material from Herbaria. In this study we have followed the indications of Wendland & Kothe (1997) and Rehner & Buckley (2005) for TEF1-α, and Frøslev et al. (2005) and Matheny (2005, 2006) for RPB2. We will take into account this recommendation of Reviewer 3 to future works in Hymenochaetaceae.
Point 16: From Table 2 legend, remove italics from ‘species’.
Response 16: It was made in the new version of the manuscript.
Point 17: In notes for Phellinotus piptadeniae, no comparison, as well as distinction with other allied species Phellinotus, was provided. I think the same will provide better clarity to the readers.
Response 17: It was made in the new version of the manuscript.
Point 18: Check for IKI- throughout the text.
Response 18: It was made in the new version of the manuscript.
Point 19: Line 579, replace ‘this taxon’ with ‘This taxon”.
Response 19: It was made in the new version of the manuscript.
Point 20: Check for gapping error on page 32.
Response 20: It was made in the new version of the manuscript.
Point 21: Line 807, check for indepented taxa.
Response 21: It was made in the new version of the manuscript.
Point 22: Please add a paragraph of conclusions based on the novelty of this kind of research and the importance of taxonomic studies to re-classify these species.
Response 22: It was made in the new version of the manuscript.
Point 23: Check for other typo and grammatical errors throughout the text.
Response 23: It was made in the new version of the manuscript.
Round 2
Reviewer 1 Report
None of my critical comments has been carefully considered and followed.
For example:
Response 2. We know these species do not belong to Phellinus, but if you do not know which genus these species belong to, how could you make combinations? You can exclude these species from Phellinus, but if you make combinations, you should provide enough evidence to support the new taxonomic position at the generic level.
Response 3. The current phylogenies are poorly constructed. The Fig. 2 does not support Phellinotus as a reliable monophyletic clade at all. You cannot say previous papers did the same thing as you, then you are right. Is it possible that previous papers are wrong? You can evidence your paper is right only by logical postulation. This is a common scientific practice. Regarding the reference Wu et al., it is under review which means the current version is not the final version. The current content will be changed. You cannot cite the current content, because it will be not present in the final version at all.
Response 5. I do not say this information is not relevant, but it should be analyzed in detail instead of just description.
The authors cannot understand what I say. They reject to improve their paper. I do not propose that the current paper should be published in any journal.